# QuantSR: Accurate Low-bit Quantization for Efficient Image Super-Resolution

**Haotong Qin**[*,1,2], **Yulun Zhang**[*,✉2], **Yifu Ding**[1], **Yifan Liu**[2],
**Xianglong Liu**[✉1], **Martin Danelljan**[2], **Fisher Yu**[2]
[1]Beihang University    [2]ETH Zürich

## Abstract

Low-bit quantization in image super-resolution (SR) has attracted copious attention in recent research due to its ability to reduce parameters and operations significantly. However, many quantized SR models suffer from accuracy degradation compared to their full-precision counterparts, especially at ultra-low bit widths (2-4 bits), limiting their practical applications. To address this issue, we propose a novel quantized image SR network, called QuantSR, which achieves accurate and efficient SR processing under low-bit quantization. To overcome the representation homogeneity caused by quantization in the network, we introduce the *Redistribution-driven Learnable Quantizer* (RLQ). This is accomplished through an inference-agnostic efficient redistribution design, which adds additional information in both forward and backward passes to improve the representation ability of quantized networks. Furthermore, to achieve flexible inference and break the upper limit of accuracy, we propose the *Depth-dynamic Quantized Architecture* (DQA). Our DQA allows for the trade-off between efficiency and accuracy during inference through weight sharing. Our comprehensive experiments show that QuantSR outperforms existing state-of-the-art quantized SR networks in terms of accuracy while also providing more competitive computational efficiency. In addition, we demonstrate the scheme's satisfactory architecture generality by providing QuantSR-C and QuantSR-T for both convolution and Transformer versions, respectively. Our code and models are released at https://github.com/htqin/QuantSR.

## 1 Introduction

Single image super-resolution (SR) is the task of obtaining a high-resolution (HR) version of a low-resolution (LR) image by retrieving high-frequency details. It is an ill-posed problem since there are multiple HR candidates for the same LR input. To address this issue, researchers have explored the use of deep neural networks, including convolutional neural networks (CNNs) and Transformers [18, 26, 38, 41, 25, 3, 39, 43], to achieve high-quality reconstruction. However, existing SR models rely on expensive computational resources, which significantly limits the real-world SR applications on resource-constrained edge devices. Therefore, there is an urgent requirement to develop model compression techniques for SR models to reduce the computational overhead.

Model quantization [5, 14, 24, 8] has emerged as a powerful compression technique that compresses weights and activations in computing units (such as convolutional and linear layers) into low-bit representations. This results in heavy floating-point operations being converted into efficient integer ones, making quantization highly desirable for edge devices and friendly for memristor-based hardware. Meanwhile, the quantization process primarily focuses on parameter compression within operations. It is considered architecture- and task-agnostic, making it popular among the deep learning

---

* Equal contribution    ✉ Corresponding authors: Yulun Zhang and Xianglong Liu

37th Conference on Neural Information Processing Systems (NeurIPS 2023).

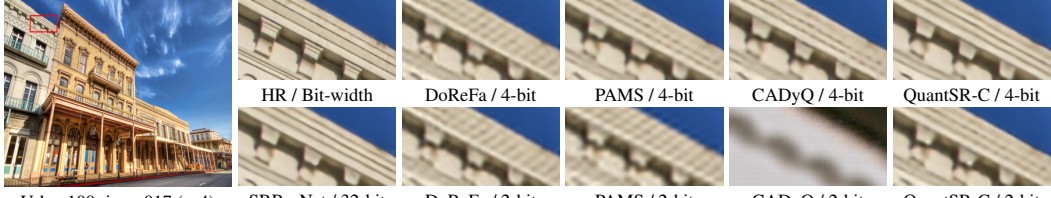

| HR / Bit-width | DoReFa / 4-bit | PAMS / 4-bit | CADyQ / 4-bit | QuantSR-C / 4-bit |
| Urban100: img_017 (×4) | SRResNet / 32-bit | DoReFa / 2-bit | PAMS / 2-bit | CADyQ / 2-bit | QuantSR-C / 2-bit |

Figure 1: Visual comparison (×4) with quantized lightweight SR models in terms of 4-bit and 2-bit. We use SRResNet [20] as the full-precision SR backbone and quantize it with low bit width. We compare our QuantSR-C with recent quantization methods (*i.e.*, DoReFa [44], PAMS [22], and CADyQ [10]). Our QuantSR-C performs obviously better than others in both 4-bit and 2-bit cases.

community. Additionally, the quantization function, known as the quantizer, can be customized for each bit width (usually 2-8 bits) to achieve a flexible balance between accuracy and efficiency.

Although quantization is known to enhance the efficiency of image SR models, it can also lead to significant performance degradation, particularly when using ultra-low bit width, *e.g.*, 2-4 bits. It should be noted that another extreme 1-bit setting (also known as binarization) [7, 28] is not considered here. This is mainly because 1-bit quantization suffers from a much larger performance gap and has a different hardware implementation in practice when compared with low-bit quantization settings. Despite attempts to minimize the loss compared to their full-precision counterparts, a considerable performance gap still exists in existing low-bit quantized SR models.

We identify two main reasons for this performance degradation. **Firstly**, the quantizer aims to compress the parameters by discretizing them. It results in the homogenization of the parameter representation in the SR model and causes the loss of gradient information during the backward pass. **Secondly**, the current quantized SR models are specific bit-width mappings of the original full-precision counterparts, and the accuracy of the latter determines its upper performance bound. And once trained, the quantized SR model loses the ability to balance accuracy and efficiency when deployed, limiting its flexibility in practice usage. In this work, we set out to address these problems, to achieve efficient, yet accurate, flexible quantized image SR models.

We propose an accurate **Quant**ized model for **S**uper-**R**esolution (**QuantSR**). We first propose a method called *Redistribution-driven Learnable Quantizer* (RLQ) that mitigates the representation homogenization caused by discretization. We achieve this by designing a redistributable and learnable quantizer that improves the forward and backward information. Our RLQ results in significant diversification of the representation without incurring additional inference burden. We further propose a *Depth-dynamic Quantized Architecture* (DQA) that advances the performance of quantized SR and provides flexibility in inference resource utilization. We construct weight-sharing deep-dynamic models from an SR architecture base with higher accuracy upper bound, enabling us to achieve superior performance and resource adaptation during inference.

Our comprehensive experiments show that QuantSR outperforms existing quantized SR models across various bit widths (see Fig. 1) by a substantial margin. Notably, our QuantSR with 4 bits achieves performance comparable to existing methods using 8 bits. We also demonstrate the effectiveness of our QuantSR on CNN- and Transformer-based SR networks, highlighting its versatility across different architectures. Our main contributions are summarized as follows:

- We propose QuantSR, a novel accurate quantization scheme for efficient image SR. It provides the potential to deploy the SR application on edge devices and further narrow the performance gap between the quantized model and its full-precision counterpart.

- We propose a Redistribution-driven Learnable Quantizer (RLQ). Specifically, our RLQ diversifies quantized representation and gradient information by redistribution in quantizers. RLQ greatly enhances representations of quantized SR models with little inference burden.

- We propose a Depth-dynamic Quantized Architecture (DQA) to achieve superior performance with the same network depth. Such a dynamic strategy further allows for multiple models with different depths, which is resource adaptation during inference.

- We employ our QuantSR to compress CNN- and Transformer- based SR networks to lower bit-width, resulting in the corresponding quantized baselines, QuantSR-C and QuantSR-T. Our QuantSR achieves superior performance over SOTA quantized SR methods.

## 2 Related Work

### 2.1 Efficient Image Super-Resolution (SR)

Recently, there has been a growing interest in efficient image super-resolution (SR) models due to their resource-friendly characteristics. Typically, researchers aim to develop lightweight networks through various methods, such as architecture design, neural architecture search (NAS), knowledge distillation (KD), and network pruning. For instance, Ahn *et al.* proposed a cascading method using a residual network [1]. While Hui *et al.* designed an information multi-distillation network [15]. Additionally, model compression techniques have also been applied to lightweight SR, such as NAS in FALSR [6], KD employed in training lighter SR student networks [9, 21], and network pruning in ASSL [42]. These techniques have obtained promising results, but they either overlook the fine-grained parameter redundancy or require dramatically additional computations, leaving significant room for compression from a bit-width perspective (*i.e.*, low-bit quantization).

### 2.2 Low-bit Quantization for SR Networks

There exist two primary methods in the quantization field: Post-Training Quantization (PTQ) and Quantization-Aware Training (QAT). PTQ has gained popularity due to its capability to quantize models without necessitating retraining. But, the absence of training and the fixed parameters in the pre-trained model constrain its potential for achieving extreme low-bit quantization [5, 17, 14, 24, 8]. Fortunately, QAT provides us with an opportunity to leverage the entire training process to achieve aggressive low-bit quantization and has shown promising performance [22, 10, 44, 4]. This method enables more comprehensive model optimization and allows for optimal training of the quantized model. QAT is thus considered powerful to achieve extremely low-bit quantization, *e.g.*, 2-4 bits.

The practical utility of existing SR networks on resource-limited devices is hindered by their extensive memory demands and computational burden. One of the significant obstacles to their usage is the extensive usage of floating-point storage and operations in these networks. As a result, there is ample opportunity to compress these networks by adopting a low-bit quantization approach, which can drastically reduce memory and computational requirements. This approach provides a compelling rationale for investigating the development of low-bit quantized SR models [22, 10, 12, 33, 16, 27]. Li *et al.* presented PAMS, which utilizes a trainable truncated parameter to dynamically determine the upper limit of the quantization range for SR models [22]. CADyQ is proposed as a technique designed for SR networks and optimizes the bit allocation for local regions and layers in the input image [10]. Hong *et al.* proposed DAQ, a channel-wise distribution-aware quantization scheme for SR models [12]. Despite these efforts, quantized SR models continue to exhibit a noticeable performance gap in comparison to their full-precision counterparts.

## 3 Method

This section first provides an overview of low-bit quantization for image super-resolution (SR) and points out the limitations of existing low-bit SR networks. We then present our **QuantSR**, an accurate **Quant**ization approach for efficient image **S**uper-**R**esolution (see Fig. 2). QuantSR mainly consists of two novel techniques: *Redistribution-driven Learnable Quantizer* (RLQ) and *Depth-dynamic Quantized Architecture* (DQA). RLQ diversifies the information of forward and backward representations. While DQA pushes the accuracy upper bound and achieves dynamic-resource inference. Finally, we discuss how to utilize QuantSR for image SR and optimize the quantized SR network. The implementation details are also provided.

### 3.1 Preliminaries

**SR Network Architecture.** We first outline the basic architecture of quantized SR networks. These networks are designed to take a low-resolution (LR) image denoted as $I_{\text{LR}}$ and produce a corresponding super-resolved image $I_{\text{SR}}$. The process can be expressed as follows

$$I_{\text{SR}} = \mathcal{M}(I_{\text{LR}}), \tag{1}$$

where $\mathcal{M}(\cdot)$ denotes the quantized super-resolution (SR) model. Specifically, the image SR network $\mathcal{M}(\cdot)$ can be generally divided into three parts: low-level feature extractor $\mathcal{E}_{\text{L}}(\cdot)$, high-level feature extractor $\mathcal{E}_{\text{H}}(\cdot)$, and reconstruction $\mathcal{R}(\cdot)$. The common practice is to apply the network quantization to the high-level feature extractor [22], which costs most of the computational resources among the

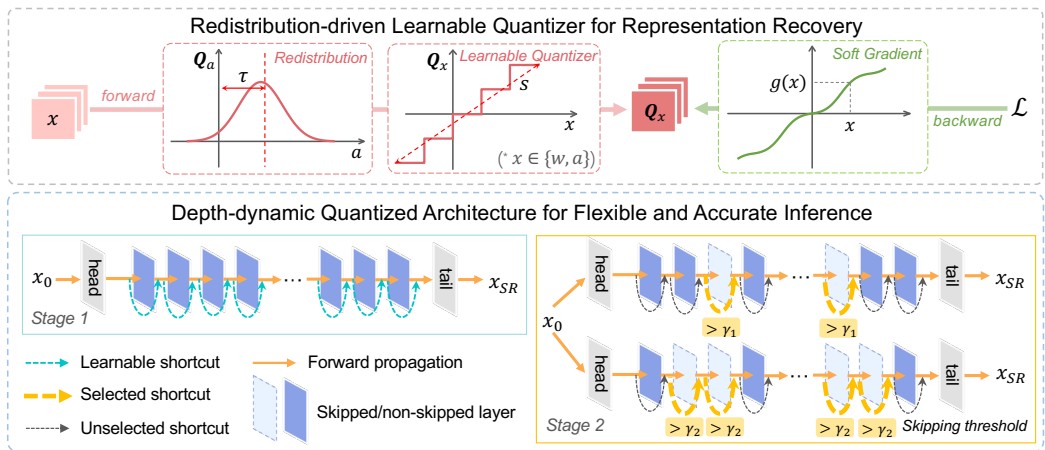

Figure 2: Overview of our QuantSR for image super-resolution network. The upper part is the *Redistribution-driven Learnable Quantizer* (RLQ), where the representation information of quantizers is enhanced in forward and backward propagation of the SR network. The lower one is the *Depth-dynamic Quantized Architecture* (DQA) that breaks the accuracy limitation of quantized SR architecture and allows for the flexible accuracy-efficiency trade-off in inference.

whole SR network. Thus, the image SR process in Eq. (1) can be further expressed as follows

$$I_{\text{SR}} = \mathcal{M}(I_{\text{LR}}) = \mathcal{R} \circ \mathcal{E}_{\text{H}} \circ \mathcal{E}_{\text{L}}(I_{\text{LR}}), \tag{2}$$

where $\mathcal{E}_{\text{H}}$ denotes the high-level feature extractor and $\circ$ denotes the connection among network parts.

**Quantization Framework.** For the quantized SR network, the weight $\boldsymbol{w}$ and activation $\boldsymbol{a}$ of computing units (such as convolution, linear, and matmul layers) are compressed to low bit-widths by the quantizers $Q_w(\boldsymbol{w})$ and $Q_a(\boldsymbol{a})$, respectively. As the common practice [35], these parameters are compressed to lower $b$ bit-widths by following quantizer $Q$ with a symmetric mode:

$$Q^b(\boldsymbol{x}) = \text{round}\left(\frac{\text{clip}(\boldsymbol{x})}{v_b}\right) v_b, \tag{3}$$

where $\boldsymbol{x}$ denotes the weight $\boldsymbol{w}$ or activation $\boldsymbol{a}$, and $Q^b(\cdot)$ denotes their quantizer. The function $\text{clip}(\cdot) = \max(\min(\boldsymbol{x}, a), -a)$ is to limit the range of the inputs, where $a$ represents the maximum of the absolute value of $\boldsymbol{x}$. And $v_b$ is the map function that scales the higher precision inputs to their lower bit reflections as $v_b = \frac{a}{2^{b-1}-1}$. Since the direct differentiation of the discrete quantizer causes all-zero gradients and hinders the backward propagation, straight-through estimation (STE) is used to approximate the gradient of parameters:

$$\frac{\partial Q^b(\boldsymbol{x})}{\partial \boldsymbol{x}} = \begin{cases} 1 & \text{if } \boldsymbol{x} \in (-a, a) \\ 0 & \text{otherwise} \end{cases}. \tag{4}$$

After quantizing the SR networks, the storage size and computation load can be significantly reduced due to the low bit-width and efficient integer operations.

## 3.2 Redistribution-driven Learnable Quantizer

### 3.2.1 Quantization-induced Representation Degradation

While quantization promises less storage and faster inference, it also causes significant performance degradation of compressed SR models. The intuitive reason is that the discretization brought by the quantizer hinders the information of representations in both forward and backward propagations. In forward propagation, the information of quantized parameters is significantly reduced due to the limitation of low bit-width, especially in ultra-low bit quantization. For example, in 2-bit quantization, the number of candidates for a single element drops from $2^{32}$ in floating point form to only 4. Furthermore, quantizers such as Eq. (3) also lead to homogeneous quantization mappings throughout the SR network. In backward propagation, since the gradient of quantizers cannot be used directly, the gradient obtained by estimators cannot completely reflect the effect of quantizers. This phenomenon results in an information mismatch always between backward and forward propagation, with the former consistently lagging behind. Therefore, the quantizer should be improved to achieve more accurate forward and backward processes, enhancing the representations of the quantized SR network.

### 3.2.2 Redistribution-driven Learnable Quantizer for Representation Recovery

In this work, we proposed a novel Redistribution-driven Learnable Quantizer (RLQ) for our QuantSR (Fig. 3). RLQ is driven by parameter redistribution designs, which introduce learnable parameters into the quantizer to diversify the forward quantization mapping in the whole quantized SR model. The transformation function is added in each quantization interval to enrich the gradient information in the backward. Our RLQ can be expressed as

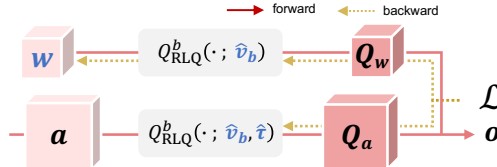

Figure 3: Forward and backward propagation of RLQ. Blue notations are learnable parameters.

$$Q_{\text{RLQ}}^b(\boldsymbol{x}, \hat{v}_b, \hat{\tau}) = \text{round}\left(\phi\left(\frac{\text{clip}(\boldsymbol{x} + \hat{\tau})}{\hat{v}_b}\right)\right)\hat{v}_b, \quad \phi(\boldsymbol{x}) = \frac{\tanh\left(2(\boldsymbol{x} - \lfloor\boldsymbol{x}\rfloor) - 1\right)}{\tanh 1} + \lfloor\boldsymbol{x}\rfloor + 2^{-1}, \quad (5)$$

where the $\hat{v}_b$ and $\hat{\tau}$ denote the learnable interval and mean-shifting parameters, respectively, which are initialized as $\frac{a}{2^{b-1}-1}$ and 0. The function $\phi(\cdot)$ is embedded within each quantization interval as a transformation function. This function does not alter the rounded value but rather reduces the gradient of elements that are distant from the interval center. The aim of this function is to incorporate the information reflecting the quantizer's actual behavior while maintaining optimization stability.

The derivative *w.r.t.* the input and learnable parameters used in the backward pass are

$$\frac{\partial Q_{\text{RLQ}}^b(\boldsymbol{x}, \hat{v}_b, \hat{\tau})}{\partial \boldsymbol{x}} = \begin{cases} \frac{\partial \phi(\boldsymbol{x} + \hat{\tau})}{\partial \boldsymbol{x}} & \text{if } \boldsymbol{x} \in (-a, a) \\ 0 & \text{otherwise} \end{cases}, \qquad \frac{\partial Q_{\text{RLQ}}^b(\boldsymbol{x}, \hat{v}_b, \hat{\tau})}{\partial \hat{\tau}} = 1 + \frac{\partial \phi(\boldsymbol{x} + \hat{\tau})}{\partial \hat{\tau}},$$

$$\frac{\partial Q_{\text{RLQ}}^b(\boldsymbol{x}, \hat{v}_b, \hat{\tau})}{\partial \hat{v}_b} = \begin{cases} \text{round}\left(\frac{\boldsymbol{x} + \hat{\tau}}{\hat{v}_b}\right) + \frac{\partial \phi((x + \hat{\tau})\hat{v}_b^{-1})}{\partial \hat{v}_b} & \text{if } \boldsymbol{x} \in (-a, a) \\ -a \text{ or } a & \text{otherwise} \end{cases}. \qquad (6)$$

Our proposed $Q_{\text{RLQ}}^b(\boldsymbol{x}, \hat{v}_b, \hat{\tau})$ quantizer is embedded in the entire quantized SR network for quantization-aware training to compress weights and activations to low bit-width.

After applying RLQ, both the forward and backward representations of QuantSR are greatly enhanced. (1) For the forward propagation, since introducing the learnable mean-shifting and quantization scale parameters to RLQ, the quantizers are gradually diversified during training. The learnable parameters of the quantizers in the entire network diversify continuously throughout the training process, which significantly reduces the impact of parameter representation degradation caused by parameter discretization. (2) For the backward propagation, the embedded transformation function $\phi(\cdot)$ can introduce information in the gradient to help accurate updates. Compared with using STE directly, RLQ provides additional gradient-guided information, that is, in each quantization interval, the farther away from the center of the region, the smaller the gradient. The transformation function is embedded in the round function and can be merged at inference time without additional burden.

### 3.3 Depth-dynamic Quantized Architecture

#### 3.3.1 Latent Architecture Bounds for Quantized SR Network

Quantization is a highly effective compression technique that employs low-bit parameters and efficient bitwise operations, making it ideal for fast inference on resource-constrained devices. One of the key benefits of quantization is that it can be applied to SR models without affecting their

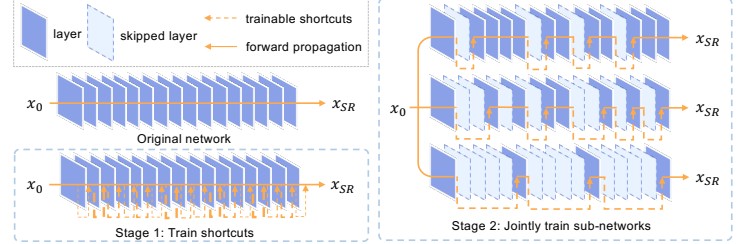

Figure 4: The architecture and training pipeline of DQA, which uses a two-stage strategy to train weight-shared variants with different depths.

underlying architecture. However, in our investigation, we observed that the quantized SR network's accuracy is typically limited by that of its full-precision counterpart and that quantization inevitably leads to some loss in accuracy. Furthermore, since SR models come in various sizes to meet different accuracy requirements, striking a balance between accuracy and efficiency becomes critical in practical applications. To address this challenge, we propose a dynamic depth quantization architecture that allows for immediate and adaptive accuracy-efficiency trade-offs in SR applications. This architecture also generates quantization models with greater accuracy at the same depth.

### 3.3.2 Depth-dynamic Quantized Architecture for Flexible and Accurate Inference

We propose a Depth-dynamic Quantized Architecture (DQA) for the quantized high-level feature extractor $\mathcal{E}'_{\mathrm{H}}$ (Fig. 4), which selects a more lightweight model with fewer layers at runtime, directly reducing the computational consumption. We first build the main quantized architecture, which is computationally expensive with learnable short connections. The main architecture $\mathcal{E}^{\mathrm{DQA}}_{\mathrm{H}}$ containing $2N$ blocks ($N \in \{2^n, n \in \mathbf{Z}^+\}$), where the initialized $i$-th block $\hat{\Phi}^{\mathrm{DQA}}_i$ is expressed as

$$\hat{\Phi}^{\mathrm{DQA}}_i(\boldsymbol{x}_i) = \varphi_i(\boldsymbol{x}_i) + \alpha_i \boldsymbol{x}_i, \tag{7}$$

where $\boldsymbol{x}_i$ denotes the input feature of $i$-th block, $\varphi(\cdot)$ denotes the quantized feature extractor consisting of convolution and activation layers, and $\alpha$ denotes the learnable scaling of the skip connection.

The training strategy of our architecture is divided into two stages. The **first stage** is a warm-up, which takes 1/5 of the training iterations. In this stage, the entire structure is optimized according to the original loss, and the skip connection is updated during the training process. In the **next stage**, the architecture is derived into a dynamic-depth version. Different derivatives are jointly trained. Specifically, the dynamic-depth architecture is allowed to be derived into different variants with the 100%, 50%, and 25% number of blocks. The variants of DQA can be expressed as

$$\mathcal{E}^{\mathrm{DQA}}_{\mathrm{H}}(x, \mathrm{var}) = \prod_{i=1}^{n} \Phi^{\mathrm{DQA}}_i(\boldsymbol{x}_i, \mathrm{var}) = \prod_{i=1}^{n} \left( b_i(\mathrm{var})\varphi_i(\boldsymbol{x}_i) + \alpha_i \boldsymbol{x}_i \right), \tag{8}$$

where $\prod$ denotes composition, $\mathrm{var} \in (0, 100\%]$ and $b_i(\mathrm{var}) \in \{0, 1\}$. The coefficient $b_i(\mathrm{var})$ of $i$-th block is determined based on the value of its scaling $\alpha_i$ of each block. The $\boldsymbol{b}[\mathrm{var}]$ is determined based on the value of its short connection $\alpha$ in each block.

For a specific variant var, we select the corresponding proportion of blocks with the smallest short connection values (meaning that the block is less likely to be skipped) after warming up and set their coefficient $b[\mathrm{var}]$ to 1. The coefficients corresponding to other blocks are set to 0 and their feature extraction processes are skipped. The reserved short connections can also be merged into the previous blocks to achieve acceleration during inference.

### 3.4 Training Pipeline of QuantSR

To optimize the SR model quantized by the proposed QuantSR, we adopt the joint training strategy to update multiple DQA variants, where the RLQ is applied to quantized weight and activation. On the given training dataset $D = \left\{ I^i_{\mathrm{LR}}, I^i_{\mathrm{HR}} \right\}^K_{i=1}$ with $K$ low-resolution inputs $I_{\mathrm{LR}}$ and corresponding high-resolution ground-truth $I_{\mathrm{HR}}$, for a specific variant var of quantized SR model $\mathcal{M}$, the conventional pixel-wise $\mathcal{L}^{\mathrm{PIX}}_{\mathrm{var}}$ loss can be expressed as

$$\mathcal{L}^{\mathrm{PIX}}_{\mathrm{var}} = \frac{1}{n} \sum_{i=1}^{k} \left\| I^i_{\mathrm{HR}} - \mathcal{M}(I^i_{\mathrm{LR}}, \mathrm{var}) \right\|_{\ell_1}. \tag{9}$$

And the losses of different variants of DQA are accumulated and used to jointly update the network:

$$\mathcal{L}_{\mathrm{tot}} = \sum_{\mathrm{var} \in \mathbf{var}} \frac{\mathcal{L}^{\mathrm{PIX}}_{\mathrm{var}}}{|\mathbf{var}|_{\ell_0}}, \quad \mathbf{var} = \{100\%, 50\%, 25\%\}, \tag{10}$$

where $|\cdot|_{\ell_0}$ denotes $\ell_0$ normalization. With the well-designed quantizer, architecture, and training pipeline, our QuantSR enjoys accuracy and efficiency with CNN- and Transformer-based backbones.

## 4 Experimental Results

### 4.1 Settings

**Dataset.** We adhere to the standard procedure in image SR, training on DIV2K [32] and evaluating on Set5 [2], Set14 [37], B100 [29], Urban100 [13], and Manga109 [30].

**Evaluation.** We report the reconstruction performance measured by PSNR and SSIM [34] on the Y channel of the YCbCr space. And we also evaluate the amounts of parameters and computation for the quantized SR model and the full-precision one.

**Proposed Quantization Baselines.** We quantize image SR models with our proposed method, including the CNN- and Transformer- based architectures. For CNN-based SR models, we follow PAMS [22] and CADyQ [10], and use SRResNet as the backbone. For Transformer-based SR models,

| Method | #Bit (w/a) | #Blk | Set5 PSNR | Set5 SSIM | Set14 PSNR | Set14 SSIM | B100 PSNR | B100 SSIM | Urban100 PSNR | Urban100 SSIM | Manga109 PSNR | Manga109 SSIM |
|---|---|---|---|---|---|---|---|---|---|---|---|---|
| SRResNet | 32/32 | 16 | 38.00 | 0.9605 | 33.59 | 0.9171 | 32.19 | 0.8997 | 32.11 | 0.9282 | 38.56 | 0.9770 |
| DoReFa | 4/4 | 16 | 37.60 | 0.9589 | 33.10 | 0.9133 | 31.87 | 0.8954 | 30.87 | 0.9151 | 37.65 | 0.9746 |
| RLQ | 4/4 | 16 | 37.72 | 0.9594 | 33.25 | 0.9147 | 31.98 | 0.8971 | 31.15 | 0.9187 | 37.99 | 0.9755 |
| DAQ | 4/4 | 32 | 37.77 | 0.9596 | 33.25 | 0.9150 | 32.00 | 0.8973 | 31.30 | 0.9200 | 38.12 | 0.9758 |
| | | 16 | 37.69 | 0.9593 | 33.15 | 0.9140 | 31.92 | 0.8963 | 30.65 | 0.9163 | 37.80 | 0.9751 |
| | | 8 | 37.51 | 0.9584 | 32.99 | 0.9122 | 31.78 | 0.8943 | 30.50 | 0.9111 | 37.38 | 0.9736 |
| QuantSR | 4/4 | 32 | 37.95 | 0.9603 | 33.59 | 0.9177 | 32.17 | 0.8996 | 31.98 | 0.9274 | 38.62 | 0.9771 |
| | | 16 | 37.88 | 0.9600 | 33.45 | 0.9166 | 32.10 | 0.8988 | 31.72 | 0.9249 | 38.37 | 0.9766 |
| | | 8 | 37.74 | 0.9595 | 33.25 | 0.9150 | 31.99 | 0.8973 | 31.27 | 0.9199 | 37.98 | 0.9757 |

Table 1: Ablation study ($\times 2$ scale) about our proposed Redistribution-driven Learnable Quantizer for Representation Recovery (RLQ) and Depth-dynamic Quantized Architecture (DQA) for flexible and accurate inference. '$w/a$' denotes the weight/activation bits. '#Blk' means residual block number.

we quantize lightweight SwinIR [25]. The implementations of comparison methods follow the official codes [11, 23] and are trained with the same settings as ours. We quantize both the weight and activation of the body part in all the models with low bit-width (*e.g.*, 2, 4, or 8). We denote $w$-bit weight and $a$-bit activation as $w/a$. To ensure a fair comparison, QuantSR reports the performance of the same-depth variant to other networks (*e.g.*, 16 blocks for SRResNet) in the comparison. The complete efficiency performance (32, 16, and 8 blocks) is demonstrated in Sec. 4.3.

**Training Strategy.** In our training process, we follow the practices of previous studies [26, 40, 36, 25] by conducting data augmentation, which involves random rotations of $90°$, $180°$, $270°$, and horizontal flipping. The models are trained for 300K iterations, with each training batch consisting of 32 image patches. The input size of each patch is $64 \times 64$. To optimize our model, we utilize the Adam optimizer [19]. The learning rate is initially set to $2 \times 10^{-4}$ and is then halved at the 250K-th iteration. All experiments are conducted on NVIDIA RTX A6000 GPUs with PyTorch [31].

## 4.2 Ablation Study

To showcase the efficacy of the techniques employed in our QuantSR, we conduct ablation studies on RLQ and DQA. We employ SRResNet [20] as the image SR backbone and trained it for 200K iterations with a 4-bit setting. To establish a quantization baseline, we used the vanilla quantization method, DoReFa [44]. Subsequently, we incorporate RLQ and/or DQA into SRResNet and binarized it. The results of our experiments are presented in Tab. 1, where we report the PSNR/SSIM values on five test benchmarks. Additionally, we provide detailed analyses of all variants (*e.g.*, 32, 16, and 8 blocks) for the depth-dynamic DQA and QuantSR.

**Redistribution-driven Learnable Quantizer**. The vanilla version of the quantized SR model, DoReFa [44], has demonstrated basic SR performance. However, we have improved upon this by incorporating RLQ for activations, which allows us to learn the shift and numerical scale. Our proposed RLQ has significantly enhanced the performance of the quantized network while reducing the performance drop. In Tab. 1, it is clear that our RLQ achieves remarkable improvements in PSNR and SSIM with little additional computation overhead, particularly with the 4-bit setting, where we achieve around 0.12-0.28 dB and 0.0005-0.0036 improvements, respectively. Our enhanced quantizer significantly improves the representation of information during both forward and backward propagation, leading to improved performance in quantized SR networks. This enhancement results in enhanced representation capabilities and more precise optimization.

**Depth-dynamic Quantized Architecture**. The proposed dynamic quantization algorithm (DQA) can adaptively operate on different versions of a model, based on the energy budget of real-world application scenarios. As shown in Tab. 1, we can use the proposed optimization scheme to jointly train models with varying degrees of quantization, where the variable in Eq. (10) can take values of 100%, 50%, and 25%. The 16-block DQA has achieved an improvement of around 0.05 dB and 0.0010 over the baseline, surpassing the upper limit of accuracy achievable by full-precision models. Notably, the DQA can effectively reduce computational complexity and memory usage by selectively ranking and skipping blocks based on a percentage threshold. Consequently, the 8-block model with the 4/4 setting has the smallest computational and storage footprint. The 32-block variant exhibits a noteworthy accuracy improvement at the cost of increased computational overhead, *e.g.*, the improvement in Set5 is 0.17 dB and 0.0005. The accuracy results of QuantSR also demonstrate that employing both RLQ and DQA techniques can enhance the performance jointly and even elevate it to full precision levels, accomplished by utilizing an efficient 4-bit width.

| Method | Scale | #Bit (w/a) | Set5 PSNR | Set5 SSIM | Set14 PSNR | Set14 SSIM | B100 PSNR | B100 SSIM | Urban100 PSNR | Urban100 SSIM | Manga109 PSNR | Manga109 SSIM |
|---|---|---|---|---|---|---|---|---|---|---|---|---|
| Bicubic | ×2 | -/- | 33.66 | 0.9299 | 30.24 | 0.8688 | 29.56 | 0.8431 | 26.88 | 0.8403 | 30.80 | 0.9339 |
| SRResNet [20] | ×2 | 32/32 | 38.00 | 0.9605 | 33.59 | 0.9171 | 32.19 | 0.8997 | 32.11 | 0.9282 | 38.56 | 0.9770 |
| SwinIR_S [25] | ×2 | 32/32 | 38.14 | 0.9611 | 33.86 | 0.9206 | 32.31 | 0.9012 | 32.76 | 0.9340 | 39.12 | 0.9783 |
| DoReFa [44] | ×2 | 8/8 | 37.32 | 0.9520 | 32.90 | 0.8680 | 31.69 | 0.8504 | 30.32 | 0.8800 | 37.01 | 0.9450 |
| CADyQ [10] | ×2 | 8/8 | 37.79 | 0.9590 | 33.37 | 0.9150 | 32.02 | 0.8980 | 31.53 | 0.9230 | 38.06 | 0.9760 |
| DoReFa [44] | ×2 | 4/4 | 37.31 | 0.9510 | 32.48 | 0.9091 | 31.64 | 0.8901 | 30.18 | 0.8780 | 36.95 | 0.9440 |
| PAMS [22] | ×2 | 4/4 | 37.67 | 0.9588 | 33.19 | 0.9146 | 31.90 | 0.8966 | 31.10 | 0.9194 | 37.62 | 0.9400 |
| CADyQ [10] | ×2 | 4/4 | 37.58 | 0.9580 | 33.14 | 0.9140 | 31.87 | 0.8960 | 30.94 | 0.9170 | 37.31 | 0.9740 |
| QuantSR-C (ours) | ×2 | 4/4 | 37.80 | 0.9597 | 33.35 | 0.9158 | 32.04 | 0.8979 | 31.46 | 0.9221 | 38.25 | 0.9762 |
| QuantSR-T (ours) | ×2 | 4/4 | 38.10 | 0.9604 | 33.65 | 0.9186 | 32.21 | 0.8998 | 32.20 | 0.9295 | 38.85 | 0.9774 |
| DoReFa [44] | ×2 | 2/2 | 36.91 | 0.9470 | 32.55 | 0.9071 | 31.41 | 0.8868 | 29.60 | 0.8740 | 36.132 | 0.9410 |
| PAMS [22] | ×2 | 2/2 | 34.04 | 0.8270 | 30.91 | 0.8751 | 30.11 | 0.8592 | 27.57 | 0.8400 | 31.79 | 0.9110 |
| CADyQ [10] | ×2 | 2/2 | 19.44 | 0.5610 | 18.51 | 0.4810 | 19.70 | 0.4760 | 17.97 | 0.4550 | 17.346 | 0.5830 |
| QuantSR-C (ours) | ×2 | 2/2 | 37.57 | 0.9589 | 33.09 | 0.9136 | 31.84 | 0.8954 | 30.77 | 0.9149 | 37.60 | 0.9745 |
| QuantSR-T (ours) | ×2 | 2/2 | 37.55 | 0.9587 | 33.12 | 0.9143 | 31.89 | 0.8958 | 30.96 | 0.9172 | 37.61 | 0.9745 |
| Bicubic | ×4 | -/- | 28.42 | 0.8104 | 26.00 | 0.7027 | 25.96 | 0.6675 | 23.14 | 0.6577 | 24.89 | 0.7866 |
| SRResNet [20] | ×4 | 32/32 | 32.16 | 0.8951 | 28.60 | 0.7822 | 27.58 | 0.7364 | 26.11 | 0.7870 | 30.46 | 0.9089 |
| SwinIR_S [25] | ×4 | 32/32 | 32.44 | 0.8976 | 28.77 | 0.7858 | 27.69 | 0.7406 | 26.47 | 0.7980 | 30.92 | 0.9151 |
| DoReFa [44] | ×4 | 4/4 | 29.57 | 0.8369 | 26.82 | 0.7352 | 26.47 | 0.6971 | 23.75 | 0.6898 | 27.89 | 0.8634 |
| PAMS [22] | ×4 | 4/4 | 31.59 | 0.8851 | 28.20 | 0.7725 | 27.32 | 0.7220 | 25.32 | 0.7624 | 28.86 | 0.8805 |
| CADyQ [10] | ×4 | 4/4 | 31.48 | 0.8830 | 28.05 | 0.7690 | 27.21 | 0.7240 | 25.09 | 0.7520 | 28.82 | 0.8840 |
| QuantSR-C (ours) | ×4 | 4/4 | 32.00 | 0.8924 | 28.50 | 0.7799 | 27.52 | 0.7342 | 25.88 | 0.7807 | 30.15 | 0.9040 |
| QuantSR-T (ours) | ×4 | 4/4 | 32.18 | 0.8941 | 28.63 | 0.7822 | 27.59 | 0.7367 | 26.11 | 0.7871 | 30.49 | 0.9087 |
| DoReFa [44] | ×4 | 2/2 | 30.54 | 0.8610 | 27.50 | 0.7538 | 26.90 | 0.7098 | 24.44 | 0.7242 | 27.31 | 0.8502 |
| PAMS [22] | ×4 | 2/2 | 29.20 | 0.8239 | 26.61 | 0.7273 | 26.36 | 0.6934 | 23.58 | 0.6812 | 25.59 | 0.8012 |
| CADyQ [10] | ×4 | 2/2 | 19.67 | 0.5380 | 19.30 | 0.4740 | 19.80 | 0.4620 | 17.97 | 0.4360 | 17.30 | 0.5640 |
| QuantSR-C (ours) | ×4 | 2/2 | 31.30 | 0.8819 | 28.08 | 0.7694 | 27.23 | 0.7246 | 25.13 | 0.7537 | 28.81 | 0.8844 |
| QuantSR-T (ours) | ×4 | 2/2 | 31.53 | 0.8845 | 28.16 | 0.7715 | 27.28 | 0.7274 | 25.26 | 0.7609 | 29.06 | 0.8898 |

Table 2: Quantitative results. SRResNet and SwinIR-S are used as full-precision backbones. '$w/a$' denotes the weight/activation bits. The best and second best results are colored with red and cyan.

## 4.3 Image Super-Resolution (SR)

We select SRResNet [20] (*i.e.*, 1,367K (×2) and 1,515K (×4) Params) and SwinIR_S [25] (*i.e.*, lightweight SwinIR with 910K (×2) and 930K (×4) Params) as CNN and Transformer backbones, respectively. The corresponding QuantSR variants are QuantSR-C and QuantSR-T, respectively. And we then compare with recent quantization methods, *e.g.*, DoReFa [44], PAMS [22], and CADyQ [10].

**Quantitative Results.** In Tab. 2, we provide PSNR and SSIM. Our 4-bit QuantSR-C achieves comparable or superior PSNR/SSIM scores than 8-bit DoReFa and CADyQ with scale ×2. In the 4-bit case, our QuantSR-C achieves 0.52 dB/0.0051 (×2) and 0.79 dB/0.0287 (×4) higher PSNR/SSIM values than CADyQ [10] on Urban100. The case of the Transformer is more challenging, as we observe a larger performance gap between the quantized QuantSR-T and the full-precision model compared to the quantized SRResNet. However, our QuantSR effectively narrows this gap, with a constant improvement compared to previous techniques, particularly at lower bit widths. For instance, with a 2-bit setting on SRResNet, our QuantSR-C outperforms PAMS by a significant 3.20 dB/0.0749 (×2) and 1.55 dB/0.0725 (×4) on Urban100. It also is significantly higher than the performance obtained with the SOTA CADyQ and approaches that of the full-precision model. Similarly, on Urban100, our QuantSR-T improves 1.26 dB/0.0125 (×2) over CADyQ in the 4-bit setting.

**Compression Ratio.** By utilizing a combination of quantization and dynamic, lightweight architecture, compression ratio of image SR models is significantly increased. In Tab. 3, we present the compression ratio and speedup in terms of Params and Ops, respectively. By quantizing the full-precision SRResNet to 2 and 4 bit-widths, we can reduce both the model size (*i.e.*, Params) and operations (*i.e.*, Ops) considerably. Following PAMS [22], we only quantize the weights and activations in the high-level feature extractor part, but we calculate the compression ratio and speedup for the entire model. With a 4-bit setting, our QuantSR-C achieves approximately 77.8% and 77.5% compression ratio for parameters and operations, respectively, through quantization alone. By integrating DQA for the 8-block variant, we were able to push the compression ratios to 83.1% and 82.9%, respectively. Furthermore, with a 2-bit setting, the operation compression ratios of 16- and 8-block variants can even reach 87.9% and 88.3%, respectively. Regarding the variant with 32

| Method | #Bit (w/a) | #Blk | Params (K) (↓ Ratio) | Ops (G) (↓ Ratio) | Urban100 PSNR | Urban100 SSIM |
|---|---|---|---|---|---|---|
| SRResNet | 32/32 | 16 | 1,367 (0%) | 90.1 (0%) | 32.16 | 0.8951 |
| QuantSR-C | 4/4 | 32 | 451 (↓ 67.0%) | 29.9 (↓ 66.9%) | 32.17 | 0.8943 |
| | | 16 | 303 (↓ 77.8%) | 20.2 (↓ 77.5%) | 32.00 | 0.8924 |
| | | 8 | 230 (↓ 83.1%) | 15.4 (↓ 82.9%) | 31.75 | 0.8894 |
| QuantSR-C | 2/2 | 32 | 170 (↓ 87.6%) | 11.5 (↓ 87.2%) | 31.48 | 0.8849 |
| | | 16 | 161 (↓ 88.2%) | 10.9 (↓ 87.9%) | 31.30 | 0.8819 |
| | | 8 | 156 (↓ 88.6%) | 10.6 (↓ 88.3%) | 31.04 | 0.8771 |

Table 3: Compression ratio of 2-bit and 4-bit SRResNet (×2), and their input sizes are 3×256×256 for calculating Ops.

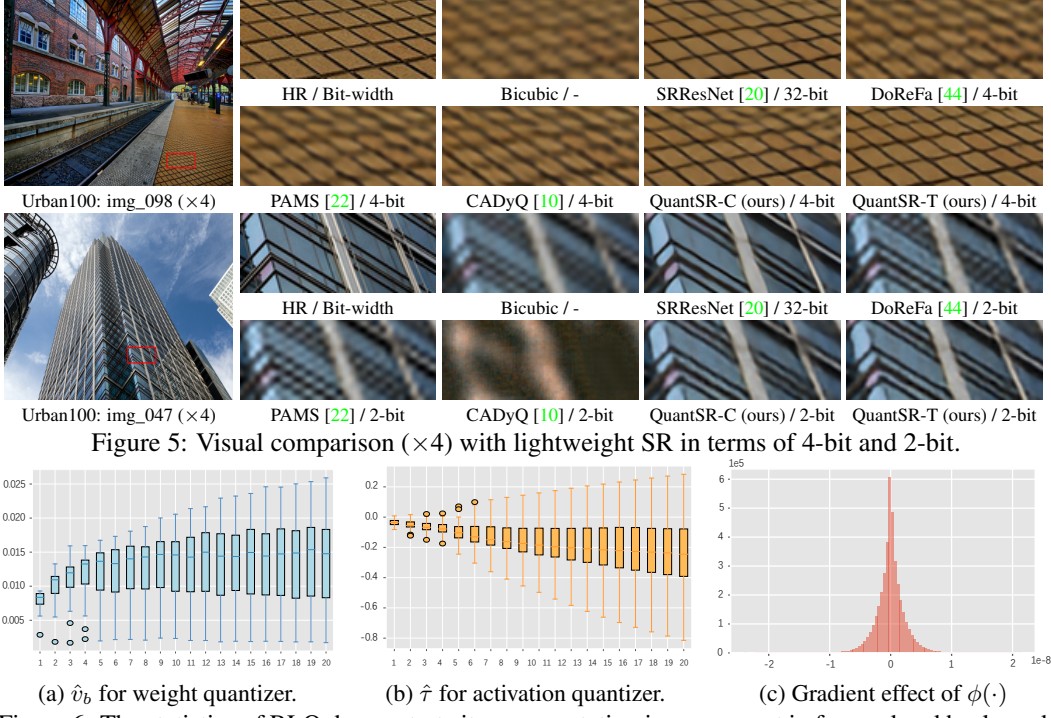

Figure 5: Visual comparison (×4) with lightweight SR in terms of 4-bit and 2-bit.

(a) $\hat{v}_b$ for weight quantizer.     (b) $\hat{\tau}$ for activation quantizer.     (c) Gradient effect of $\phi(\cdot)$

Figure 6: The statistics of RLQ demonstrate its representation improvement in forward and backward.

blocks, 4-bit QuantSR-C model also achieves notable parameter and operation savings of 67.0% and 66.9%, respectively, while obtaining comparable accuracy with the 16-block full-precision model.

## 4.4 Visualization

**Visual Results.** In Fig. 5, we provide visual results of representative methods with scale ×4 in terms of 2-bit and 4-bit cases. For each case, we compare with several quantization methods, like DoReFa [44], PAMS [22], and CADyQ [10]. Our QuantSR-C and QuantSR-T recover more structural details and alleviate more blurring artifacts than other quantization methods. Consequently, the visual difference between our QuantSR and the full-precision one is small. These visual comparisons further demonstrate the effectiveness of our QuantSR, which is consistent with the observations in Tab. 2.

**Training Statistics.** In Fig. 6, we present statistics of the learnable parameters and function effects in our RLQ for every 100 iterations of training. Specifically, we report the statistics of $\hat{v}_b$, which represents the activations, $\hat{\tau}$, which represents the weights, and the gradient effects of $\phi(\cdot)$. Figures 6 (a) and (b) illustrate that the learnable parameters are initially initialized in close proximity to each other. However, as training progresses, the optimized parameters diverge in various directions. This indicates that the quantizers containing these parameters in QuantSR are becoming more diverse, thus recovering the lost representation information due to discretization. In Fig. 6 (c), we depict the gradient change induced by a specific embedded transformation function $\phi(\cdot)$ during backpropagation. While the forward results and training stability remain unaffected, the gradient guidance information provided by this function is significant. Consequently, it promotes the optimization of QuantSR, leading to an overall improvement in the results.

## 5 Conclusion

In this paper, we propose QuantSR, a novel quantized image super-resolution (SR) network that overcomes the limitations of existing low-bit quantized SR models. The proposed network employs a Redistribution-driven Learnable Quantizer (RLQ) to improve the representation ability of quantized networks and a Depth-dynamic Quantized Architecture (DQA) to achieve flexible inference and break the upper limit of accuracy. The authors demonstrate that QuantSR outperforms existing state-of-the-art quantized SR networks in both accuracy and computational efficiency. Additionally, we provide QuantSR-C and QuantSR-T for both convolution and Transformer versions, respectively, to demonstrate the scheme's satisfactory architecture generality.

**Acknowledgements.** This work was supported by the National Natural Science Foundation of China (No. 62022009), the State Key Laboratory of Software Development Environment (SKLSDE-2022ZX-23), and the Huawei Technologies Oy (Finland) Project.

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
