# OpenReview forum: "QuantSR: Accurate Low-bit Quantization for Efficient Image Super-Resolution"
_NeurIPS.cc/2023/Conference — NeurIPS 2023 spotlight_

### Official Review · Reviewer_otEX · 2023-06-27

**Soundness:** 3 good
**Presentation:** 3 good
**Contribution:** 4 excellent
**Rating:** 8
**Confidence:** 5

**Summary:**

This manuscript is devoted to pushing the super-resolution (SR) models to ultra-low bit-width (2-4 bits). It proposes two methods and pushes the bit-width to ultra-low 2/4 bits with little accuracy loss. Meanwhile, the proposed methods not only boost the accuracy performance but also reduces the parameters and computation. It is rather difficult since the fewer parameters bring limited representation capability. The methods are well-motivated and effective, clearly demonstrating their motivation.

**Strengths:**

There are several strengths here:
(1) This work proposes two methods; the redistribution-driven learnable quantizer (RLQ) mainly helps to improve the accuracy performance with little computation overhead, while the Depth-dynamic quantized architecture (DQA) further reduces the computation and parameters by integrating structured pruning. It is an interesting attempt to combine learnable structured pruning with learnable low-bit quantization.

(2) Results shown in tables and visualizations are impressive. Figure 1 compares several quantization methods, and QuantSR shows a dominant advantage that can be clearly recognized. Many recently proposed methods are listed and compared, while QuantSR achieves consistent performance under 4/2 bit-width.

(3) First, the motivation is clear: SR models rely on expensive computational resources, and model quantization is an effective approach that can reduce the model parameters and computation. Second, the equations are correct, and notations are well demonstrated, and the figures are helpful in understanding. Moreover, the references are extensive and cited in the correct places.

(4) Compared with existing quantized SR methods, QuantSR achieves higher accuracy performance under 2/4 bit-width, which is close to the full-precision methods. And the compression ratio shown in Table 3 also shows the potential of quantization for practical value. Both quantizing and block dropping save the parameters and operations while keeping the accuracy. It can promote the implementation of quantizing SR networks in real-world scenarios.

**Weaknesses:**

There are several weaknesses here:
(1) The functions in RLQ should be further clarified. Firstly, during the forward propagation, RLQ includes the function \phi, but it is not clear whether it plays a role in the forward. The author needs to clarify its function and discuss whether it would reduce the inference efficiency of the quantized model. Secondly, I am confused about the three equations in the backward propagation. As I understand, the function \phi primarily affects x (the input to the quantization function) in the backward, as indicated in the first equation of Eq 6. I suggest the author further discuss the meaning of these Eqs.

(2) The results among CNNs and transformers are strong, but the latter seems to lack comparison, I suggest authors implement SOTA quantized SR method to transformer-based SR networks and report their results, it would be significantly helpful to improve the experiment part.

(3) The writing of the paper should be improved, there are grammatical errors in the existing manuscript, and some symbols are not clarified. The authors should correct them thoroughly in the next revision.

**Questions:**

The authors should respond to the raised issues in the weakness part.

(1) Here is a concern about the RLQ. As far as I know, the LSQ quantization method [1] has been proposed to learn quantization steps for better projecting the floating-point values to the integer space. So can you clarify the difference between the RLQ in QuantSR and the existing quantization method?
[1] LEARNED STEP SIZE QUANTIZATION. ICLR, 2021.

(2) I notice that there are more learnable parts introduced to QuantSR, including the learnable interval and mean-shifting parameters in RLQ, and the trainable shortcut scores in DQA. I want to know the training difficulty of the proposed methods—any training algorithms/tricks and detailed strategies that are used in the training framework.

**Limitations:**

See weaknesses and questions parts.

---

> ### Author Rebuttal · Authors · 2023-08-09
>
> > **Q1a**: The functions in RLQ should be further clarified. …, but it is not clear whether it plays a role in the forward. The author needs to clarify its function and discuss whether it would reduce the inference efficiency of the quantized model.
>
> **A1a**: We clarify that the $\phi$ function can be accurately embedded in the quantization interval of RLQ without affecting the quantized result in forward, and introducing information into the gradient when deriving backwards. Therefore, the $\phi$ function **does not affect the inference speed** of the quantized SR model, but only improves the optimization during training.
>
> > **Q1b**: Secondly, I am confused about the three equations in the backward propagation. …
>
> **A1b**: We clarify that Eq. (6) presents the derivative functions of the learnable parameters in the backward propagation. These include the derivative function for the input $\mathbf x$ (weights $\mathbf w$ or activation $\mathbf a$) (first equation), and the derivative function for the zero point $\tau$ (second equation), and the derivative function with respect to the quantization interval $v$ (third equation). In these derivative functions, the influence of the embedding function $\phi$ is considered and thus introduces information into the gradient to improve training.
>
> > **Q2**: The results among CNNs and transformers are strong, but the latter seems to lack comparison, …
>
> **A2**: We follow the reviewer's suggestion to compare more quantized transformer-based SR models. Specifically, we demonstrate the results of 4-bit SwinIR and CAT quantized by DoReFa and PAMS in Table A2 (*Table 1 of the attached PDF*). Our QuantSR method is able to stably outperform existing methods on different transformer-based SR models, implying that the advantages of the proposed techniques of QuantSR are robust across different architectures.
>
> Table A3a: Quantitative results. SwinIR and CAT-R are used as full-precision backbones.
> |                             |                         | \#Bit   | Set5 |  | Set14 | | B100 | | Urban100 | | Manga109 | |
> |-----------------------------|-------------------------|-----------|-------|--------|-------|--------|-------|--------|-------|--------|-------|-------------|
> | Method    | Scale | ($w$/$a$) | PSNR  | SSIM   | PSNR  | SSIM   | PSNR  | SSIM   | PSNR  | SSIM   | PSNR  | SSIM |
> | Bicubic   | $\times$4  | -/-   | 28.42                  | 0.8104                  | 26.00                  | 0.7027                  | 25.96                  | 0.6675                  | 23.14                  | 0.6577                  | 24.89                  | 0.7866                  |
> | SwinIR\_S | $\times$4 | 32/32 | 32.44                  | 0.8976                  | 28.77                  | 0.7858                  | 27.69                  | 0.7406                  | 26.47                  | 0.7980                  | 30.92                  | 0.9151                  |
> | DoRaFa    | $\times$4 | 4/4   | 29.65                  | 0.8416                  | 26.92                  | 0.7383                  | 26.53                  | 0.6993                  | 23.85                  | 0.6957                  | 26.15                  | 0.8218                  |
> | PAMS      | $\times$4 | 4/4   | 31.52                  | 0.8865                  | 28.19                  | 0.7727                  | 27.33                  | 0.7279                  | 25.35                  | 0.7620                  | 29.24                  | 0.8924                  |
> | QuantSR-T | $\times$4 | 4/4   | 32.18 | 0.8941 | 28.63 | 0.7822 | 27.59 | 0.7367 | 26.11 | 0.7871 | 30.49 | 0.9087 |
> | Bicubic                     | $	\times$4                | -/-       | 28.42 | 0.8104 | 26.00 | 0.7027 | 25.96 | 0.6675 | 23.14 | 0.6577 | 24.89 | 0.7866      |
> | CAT\_R | $\times$4               | 32/32     | 32.89 | 0.9044 | 29.13 | 0.7955 | 27.95 | 0.7500 | 27.62 | 0.8292 | 32.16 | 0.9269      |
> | PAMS                        | $\times$4               | 4/4       | 31.77 | 0.8885 | 28.33 | 0.7752 | 27.40 | 0.7295 | 25.55 | 0.7672 | 29.62 | 0.8966      |
> | QuantSR-T                   | $\times$4               | 4/4       |  32.31 | 0.8955 | 28.74 | 0.7836 | 27.61 | 0.7389 | 26.31 | 0.7896 | 30.69 | 0.9118     |
>
> > **Q3**: The writing of the paper should be improved, …
>
> **A3**: We will follow your suggestions to carefully refine our manuscript in the final revision.
>
> > **Q4**: … can you clarify the difference between the RLQ in QuantSR and the existing quantization method (LSQ [1])?
>
> **A4**: Compared with LSQ mentioned by the reviewer, our RLQ comprehensively improves the forward and backward of the quantizer to enhance the propagated information in the quantized SR network.
>
> Specifically, for forward propagation, RLQ not only learns the quantization interval but also the zero-point of the quantizer, allowing for significant diversification of the quantized parameter in the whole network. For backward propagation, the soft embedded function is to incorporate information reflecting the quantizer's actual behavior while ensuring optimization stability. These improvements make RLQ significantly outperform the existing quantizer.
>
> > **Q5**: … I want to know the training difficulty of the proposed methods—any training algorithms/tricks and detailed strategies that are used in the training framework.
>
> **A5**: The training of QuantSR exhibits stability and robustness. We provide a comprehensive explanation of our training pipeline and experimental setup in Sec 3.4 and 4.1, respectively. The training of all learnable parameters is uniformly performed using the loss function defined by Eq. (10), without any additional constraints.

---

> > ### Comment · Reviewer_otEX · 2023-08-19
> > **Response to Authors**
> >
> > Thanks for the authors' reply. The rebuttal addressed my concerns well. After carefully reading other reviews and the rebuttals, I would keep my rating for the sufficient contributions and well-supported experiments. Although there are several quant methods in high-level tasks, while there are still few explorations on low-level tasks. Thus, this paper would be helpful for the future work in light-weight low level vision.

---

### Official Review · Reviewer_j1Cc · 2023-06-28

**Soundness:** 3 good
**Presentation:** 4 excellent
**Contribution:** 3 good
**Rating:** 7
**Confidence:** 5

**Summary:**

This paper proposes a novel quantization method for SR models, including two new technologies to improve the unit representation and architectural potential of quantized SR models and push low-bit SR models to full-precision performance. The results on CNN and Transformer models on SR tasks show that QuantSR achieves SOTA performance and exceeds existing SR quantization methods. The most significant point of this paper is that the dynamic quantized architecture of QuantSR pushes up the performance ceiling of quantized SR from a block-stacking perspective and allows flexible reasoning according to resources in actual deployment, leading to a new promising way to lossless quantized SR.

**Strengths:**

In this paper, the authors propose novel RLQ and DQA technologies to jointly improve quantized SR models from two orthogonal perspectives of the unit representation and architectural potential:

a) RLQ is an effective technique. By introducing learnable parameters and redistribution functions in the forward and backward propagation of quantized computing units, the amount of information in the quantization training propagation process is significantly improved and more accurate, and the representation ability of the quantization unit is improved.

b) DQA is more attractive to me. In addition to being effective, this is a novel attempt to improve the quantized SR model from an architectural perspective. QuantSR's dynamic quantization architecture pushes up the performance ceiling of quantized SR from a block-stacking perspective, thus providing a new promising way towards lossless quantized SR models. In addition, it also allows the quantized SR model to perform flexible reasoning according to the resources in the actual deployment, which means that the practicality of the quantized SR model is greatly enhanced.

The experimental results are superior. QuantSR shows significant performance improvement under all bit widths and surpasses the existing quantized SR methods. This enables QuantSR to achieve a SOTA structure while being novel. The visualization results clearly show that the effect of the proposed technologies follows their motivation.


**Weaknesses:**

a) One notable issue is there are some places in the paper that should be revised and clarified, including but not limited to the following:
- In the box of Redistribution in Fig.2, should the y-axis coordinate be PDF(a), that is, the probability density function of a
- In the box of Learnable Quantizer in Fig.2, according to the formula Eq.5 in the paper, s should be corrected to vˆb
- The gamma in Stage 2 in Fig.2 is undefined
- Symbol a is defined repeatedly as activation or range in Eq.4
- According to the implementation in the article, the options of var may be actually 200% (32 blocks), 100% (16 blocks), and 50% (8 blocks)

b) In RLQ, the ϕ function seems to be used as an estimator to replace STE, so why it needs to be used in the forward pass and not just backward? In addition, the authors also need to explain why the ϕ function adopts this shape instead of the existing soft quantization functions such as DSQ [1] and Bireal [2].

c) The author should compare more quantized transformer SR models, and existing experiments only show the QuantSR performance on this type of network, other quantized methods are also needed. And it is necessary to show the comparison of the parameters and calculation amount of SwinIR and SRResNet, including full precision and quantized versions, to make the comparison more fair and clear.

[1] Gong R, Liu X, Jiang S, et al. Differentiable soft quantization: Bridging full-precision and low-bit neural networks, ICCV, 2019.
[2] Liu Z, Wu B, Luo W, et al. Bi-real net: Enhancing the performance of 1-bit cnns with improved representational capability and advanced training algorithm, ECCV, 2018.


**Questions:**

See weaknesses.

**Limitations:**

The author proposes an effective quantification method for SR tasks in this paper, and I suggest the author discuss the potential of QuantSR in more low-level vision tasks in more detail, which will make the contributions of the proposed method more significant and wide.

---

> ### Author Rebuttal · Authors · 2023-08-09
>
> > **Q1**: One notable issue is there are some places in the paper that should be revised and clarified, including but not limited to the following: [...]
>
> **A1**: Thank you for pointing them out. We will carefully revise them in our final revision.
>
> > **Q2a**: In RLQ, …, so why it needs to be used in the forward pass and not just backward?
>
> A2a: We clarify that the $\phi$ function can be accurately embedded in the quantization interval of RLQ without affecting the quantized result and introducing information into the gradient when deriving backwards. Therefore, the $\phi$ function does not affect the inference speed of the quantized SR model, but only improves the optimization during training.
>
> > **Q2b**: In addition, … why the ϕ function adopts this shape instead of the existing soft quantization functions such as DSQ [1] and Bireal [2].
>
> **A2b**: Compared to existing quantization methods that use soft functions, such as the tanh-based DSQ function [1] and quadratic-based function [2], the $\phi$ function in RLQ has a stable impact in once backward while the former two are more intense. This property allows the quantized SR model to update stably without collapsing and ultimately enhances the accuracy by introducing forward vectorized information into the gradients. The results in Table 4 demonstrate that QuantSR significantly outperforms other SR model quantized soft-function-based methods.
>
> > **Q3a**: The author should compare more quantized transformer SR models, and existing experiments only show the QuantSR performance on this type of network, other quantized methods are also needed.
>
> **A3a**: We follow the reviewer's suggestion to compare more quantized transformer-based SR models. Specifically, we demonstrate the results of 4-bit SwinIR and CAT quantized by DoReFa and PAMS in Table A3a (*Table 1 of the attached PDF*). Our QuantSR method is able to stably outperform existing methods on different transformer-based SR models, implying that the advantages of the proposed techniques of QuantSR are robust across different architectures.
>
> Table A3a: Quantitative results. SwinIR and CAT-R are used as full-precision backbones.
> |                             |                         | \#Bit   | Set5 |  | Set14 | | B100 | | Urban100 | | Manga109 | |
> |-----------------------------|-------------------------|-----------|-------|--------|-------|--------|-------|--------|-------|--------|-------|-------------|
> | Method    | Scale | ($w$/$a$) | PSNR  | SSIM   | PSNR  | SSIM   | PSNR  | SSIM   | PSNR  | SSIM   | PSNR  | SSIM |
> | Bicubic   | $\times$4  | -/-   | 28.42                  | 0.8104                  | 26.00                  | 0.7027                  | 25.96                  | 0.6675                  | 23.14                  | 0.6577                  | 24.89                  | 0.7866                  |
> | SwinIR\_S | $\times$4 | 32/32 | 32.44                  | 0.8976                  | 28.77                  | 0.7858                  | 27.69                  | 0.7406                  | 26.47                  | 0.7980                  | 30.92                  | 0.9151                  |
> | DoRaFa    | $\times$4 | 4/4   | 29.65                  | 0.8416                  | 26.92                  | 0.7383                  | 26.53                  | 0.6993                  | 23.85                  | 0.6957                  | 26.15                  | 0.8218                  |
> | PAMS      | $\times$4 | 4/4   | 31.52                  | 0.8865                  | 28.19                  | 0.7727                  | 27.33                  | 0.7279                  | 25.35                  | 0.7620                  | 29.24                  | 0.8924                  |
> | QuantSR-T | $\times$4 | 4/4   | 32.18 | 0.8941 | 28.63 | 0.7822 | 27.59 | 0.7367 | 26.11 | 0.7871 | 30.49 | 0.9087 |
> | Bicubic                     | $	\times$4                | -/-       | 28.42 | 0.8104 | 26.00 | 0.7027 | 25.96 | 0.6675 | 23.14 | 0.6577 | 24.89 | 0.7866      |
> | CAT\_R | $\times$4               | 32/32     | 32.89 | 0.9044 | 29.13 | 0.7955 | 27.95 | 0.7500 | 27.62 | 0.8292 | 32.16 | 0.9269      |
> | PAMS                        | $\times$4               | 4/4       | 31.77 | 0.8885 | 28.33 | 0.7752 | 27.40 | 0.7295 | 25.55 | 0.7672 | 29.62 | 0.8966      |
> | QuantSR-T                   | $\times$4               | 4/4       |  32.31 | 0.8955 | 28.74 | 0.7836 | 27.61 | 0.7389 | 26.31 | 0.7896 | 30.69 | 0.9118     |
>
> > **Q3b**: And it is necessary to show the comparison of the parameters and calculation amount of SwinIR and SRResNet, including full precision and quantized versions, to make the comparison more fair and clear.
>
> **A3b**: We further present the FLOPs and storage of the full-precision and quantized transformer-based SwinIR models, and compare them with SRResNet and QuantSR-C. As shown in Table A3b, QuantSR can also bring significant speedup and compression to transformer-based models, which shows that QuantSR brings general efficiency improvements on different architectures.
>
> Table A3b. Comparison of the number of parameters and operations.
> |  Method    | \#Block     | \#Bit($w$/$a$) | \#Params (M) | Ops (G) |
> |-------------|-----------|----------------|--------------|-----------|
> |  SRResNet  | 16  | 32/32          | 1.37        | 90.1      |
> |  QuantSR-C | 16 | 4/4            | 0.30      |     20.2      |
> |  SwinIR-S  | 4  | 32/32          | 0.88 | 195.6      |
> |  QuantSR-T | 4 | 4/4            | 0.20      |     43.4      |

---

> > ### Comment · Reviewer_j1Cc · 2023-08-16
> > **Thank you for addressing my comments**
> >
> > I appreciate your efforts in conducting additional experiments and incorporating my suggestions. The paper was a pleasure to read and I am maintaining my score of 7. Thank you.

---

### Official Review · Reviewer_E4FQ · 2023-07-01

**Soundness:** 3 good
**Presentation:** 3 good
**Contribution:** 3 good
**Rating:** 5
**Confidence:** 4

**Summary:**

The paper proposes a new quantization scheme for super-resolution (SR) networks. The paper starts with a claim that weight quantization results in the homogeneity of parameters, leading to the loss of gradient information during backward pass. To resolve the claimed issue, the paper introduces Redistribution-driven Learnable Quantizer (RLQ) to reduce homogenization that is caused by discretization. RLQ is implemented by redistributable and learnable quantizer that improves the quality of information during both forward and backward passes, resulting in diverse representation, without additional inference overhead. The paper further introduces a Depth-dynamic Quantized Architecture (DQA) to better handle the performance-efficiency trade-off through learnable short-cut connections and weight sharing from different networks.

**Strengths:**

- The paper is easy to read and well-written
- The paper introduces a new method RLQ to handle weight homogeneity issues, without additional inference overhead.
- The paper introduces DQA to better control the trade-off between performance and efficiency.

**Weaknesses:**

- Missing discussions and quantitative comparisons against related works. What is the major difference between DQA and DropConnect [A] and AIG [B]? Also, why limiting to variants 100%, 50%, and 25% in comparison to these two related works?

---

- The paper claims that there exists weight homogeneity due to quantization, which the proposed method mitigates. However, the paper does not present the demonstration (neither statistics nor visualization) of the extent of weight homogeneity before and after the proposed quantization scheme.

---

- Can the authors provide intuitive explanation as to how the proposed method mitigates the weight homogeneity issues? Isn't the proposed method still susceptible to weight homogeneity, due to the limited possible number of values each weight can take? I'm guessing the learnable mean-shifting and quantization scale parameters are optimized to diversify weight values in the given possible values each weight can take. But, I can't see how the objective function and the current design can guarantee such diversification. More intuitive and detailed explanations could be helpful.

---

- Can the authors provide intuitive explanation as to how the proposed method mitigates the weight homogeneity issues? Isn't the proposed method still susceptible to weight homogeneity, due to the limited possible number of values each weight can take?

---

- Last but not least, there is no latency/bit operation efficiency comparisons against related works, such as PAMS.


---

[A] Regularization of Neural Networks using DropConnect \
[B] Convolutional Networks with Adaptive Inference Graphs

**Questions:**

Refer to weakness section.

**Limitations:**

The authors have addressed the limitations in the supplementary section.

---

> ### Author Rebuttal · Authors · 2023-08-09
>
> > **Q1**: Missing discussions and quantitative comparisons against related works. What is the major difference between DQA and DropConnect [A] and AIG [B]? Also, why limiting to variants 100%, 50%, and 25% in comparison to these two related works?
>
> **A1**: We will incorporate the related work suggested by the reviewer.
>
> Compared to these dynamic networks, DQA employs low-bit parameters and undergoes quantization-aware training. It is more significant that DQA is mainly devoted to utilizing a dynamic architecture to break through the precision limitation of quantized models. Benefiting from screening from a stronger initial SR model with more stacked quantized blocks, the upper boundary of the model performance with the original block size is shattered, resulting in enhanced accuracy. This means that the architecture not only allows quantized SR networks to achieve dynamic trade-offs during inference but also significantly improves the accuracy at the same model size.
>
> Due to the joint training of various variants, we just use three variants for QuantSR to control the training costs. For example, for 16-block SRResNet, QuantSR simultaneously trains weight-shared quantized variants with 32, 16, and 8 blocks, where the first and last one strike a tilt in accuracy and efficiency, respectively. It is worth noting that our method, based on the stacking of blocks, thus also allows for a more fine-grained selection of variants with varying the number of blocks based on specific requirements.
>
> > **Q2**: The paper claims that there exists weight homogeneity due to quantization, which the proposed method mitigates. However, the paper does not present the demonstration (neither statistics nor visualization) of the extent of weight homogeneity before and after the proposed quantization scheme.
>
> **A2**: Our RLQ achieves diversification of quantizers throughout the network during training by utilizing learnable intervals and zero points. Fig. 6(a) in the manuscript demonstrates that the learnable parameters of the weight quantizers gradually diversify with increasing training epochs. This indicates that the discretized mapping of weights in different layers starts to vary, resulting in diversified quantized weights across the entire network.
>
> Following your suggestion, we further present a comparison between visualizations of quantized weights with and without trainable parameters (see *Fig. 1 in the attached PDF*). As observed, the presence of learnable parameters in QuantSR leads to a progressive diversification of quantizers, consequently resulting in gradually diversified quantized weights across the entire network. In the quantized SR network without learnable parameters, the weights are more homogenized during training, and their distribution remains nearly unchanged. This illustrates the significant role of RLQ in addressing weight homogeneity.
>
> Overall, these findings demonstrate the effectiveness of RLQ in promoting diversity among quantizers and mitigating weight homogenization in the quantization process.
>
> > **Q3**: Can the authors provide an intuitive explanation as to how the proposed method mitigates the weight homogeneity issues? …
>
> **A3**: Our RLQ achieves the diversification of the quantizer by utilizing learnable intervals and zero points, guiding the diversified expression of quantized weights throughout QuantSR. While the quantization interval for each quantizer remains unchanged (e.g., 2^4 for 4-bit), the quantized weights exhibit diverse trends due to different quantizers learning distinct mappings during the training process. Detailed discussions and visualizations are presented in A2.
>
> Furthermore, it is worth noting that our RLQ does not require additional constraints and can be directly optimized using the original loss function. Since discretization leads to the degradation of pixel-level information in the SR network, introducing learnable parameters into the quantizer naturally promotes the diversification of both weights and quantizers without the need for extra objectives. This point is demonstrated by the visualizations in Fig. 6 of our original paper and *Fig. 1 in the attached PDF*.
>
> > **Q4**: Last but not least, there is no latency/bit operation efficiency comparisons against related works, such as PAMS.
>
> **A4**: We will compare the efficiency with other methods in the paper as you suggested. Since our techniques do not introduce any additional computation on inference, QuantSR has the same efficiency as other quantized SR models with significantly improved accuracy under the architecture with the same number of blocks. We present the comparison in Table A4.
>
> Table A4. Comparison of the number of parameters and operations.
> |  Method    | \#Block     | \#Bit($w$/$a$) | \#Params (M) | Ops (G) |
> |-------------|-----------|----------------|--------------|-----------|
> |  SRResNet  | 16  | 32/32          | 1.37        | 90.1      |
> |  DoReFa    | 16 | 4/4            | 0.30         | 20.2      |
> |  DSQ       | 16 | 4/4            | 0.30         |      20.2     |
> |  Bi-Real   | 16 | 4/4            | 0.30         | 20.2      |
> |  PAMS      | 16 | 4/4            | 0.30         | 20.2      |
> |  CADyQ     | 16 | 4/4            | 0.30         | 20.2      |
> |  QuantSR-C | 16 | 4/4            | 0.30      |     20.2      |

---

> > ### Comment · Reviewer_E4FQ · 2023-08-18
> >
> > I would like to thank the authors for the rebuttal.
> > Some of my concerns still remain after carefully reading the rebuttal and going over the paper and related works again.
> > 1. Efficiency: I'm interested in seeing inference latency comparisons as noted in the original review. I'm curious as to how the latency compares against baselines to actually observe if there is overhead or not.
> >
> > 2. Related work performance:
> > - One of recent works (with SOTA performance) is not included:
> >     - Dynamic Dual Trainable Bounds for Ultra-low Precision Super-Resolution Networks, ECCV 2022.
> >     - Can authors provide discussions and performance comparisons against the work?
> > - There are huge discrepancies between the reported performance of baselines for SRResNet provided in the paper and DDTB (related work mentioned above). Why is that?
> > - How did authors reproduce results of CADyQ with fixed bit width for weights and activations, when CADyQ dynamically sets bits for each image?

---

> > > ### Author Response · Authors · 2023-08-18
> > >
> > > We deeply appreciate the reviewer's feedback and answer the questions as follows:
> > >
> > > > **Q1**: Efficiency: I'm interested in seeing inference latency comparisons as noted in the original review. I'm curious as to how the latency compares against baselines to actually observe if there is overhead or not.
> > >
> > > **A1**: We clarify that the latency of the quantized SR models on real deployment depends on the specific deployment library and hardware [1] (e.g., the INT8 quantized model has about 3x speedup compared to the FP32 counterpart on batch 1 setting on Tesla T4 GPU with TensorRT). It is hard to implement and deploy all quantized SR models on real hardware in such a limited time, while we plan to evaluate their latency when deployed in future work following the reviewer's suggestion.
> > >
> > > Moreover, we highlight that under the same architecture, QuantSR does not bring additional computational burden. As Table A4 shows in our last response, a comprehensive comparison of computational FLOPs reveals that our proposed QuantSR did not bring additional FLOPs while bringing improved accuracy. Therefore, on real deployment, the latency gap with other quantitative SR models may be minor.
> > >
> > > [1] MQBench: Towards Reproducible and Deployable Model Quantization Benchmark. Gong, et al. NeurIPS, 2021.
> > >
> > > > **Q2a**: Related work performance: One of recent works (with SOTA performance) is not included: DDTB. Can authors provide discussions and performance comparisons against the work?
> > >
> > > **A2a**: We compare DDBT in Table A2a and Table 2 of the attached PDF, and also discuss and show that the RLQ in our QuantSR can well solve the high-dynamic activation problem proposed by [2] (as in A7 for Reviewer G4U2). We will include relevant comparisons and discussions in our next version.
> > >
> > > As we demonstrate in the following Table A2a and Table 2 of the attached PDF, QuantSR consistently outperforms the DDTB method on various datasets. Moreover, in our QuantSR, the learnable parameters in RLQ effectively tackle the quantization problem of high dynamic activations in SR networks. Since the interval and zero-point of RLQ are learnable, it can gradually adapt to diverse and asymmetric input distributions. This implies that it can handle high dynamic activations better than statistics-based quantizers. As shown in Fig. 6(b), with increasing training, the offset of the activation quantizer in the network becomes more diverse, confirming the activation in the SR network is highly dynamic and demonstrating the ability of RLQ to adapt and cope well with such cases during training.
> > >
> > > [2] Dynamic Dual Trainable Bounds for Ultra-low Precision Super-Resolution Networks. Zhong, et al. ECCV, 2022.
> > >
> > > Table A2a. Quantitative results (4-bit). SRResNet is used as full-precision backbones.
> > > |  |  | \#Bit   | Set5 |  | Set14 | | B100 | | Urban100 | | Manga109 | |
> > > |--|--|--|--|--|--|--|--|--|--|--|--|--|
> > > | Method | Scale | ($w$/$a$) | PSNR    | SSIM     | PSNR    | SSIM     | PSNR    | SSIM     | PSNR    | SSIM     | PSNR    | SSIM     |
> > > | DDTB            | $\times$2 | 4/4   | 37.78 | 0.9600 | 33.32 | 0.9160 | 32.03 | 0.8980 | 31.40 | 0.9210 | -- | -- |
> > > | QuantSR-C | $\times$2 | 4/4   | 37.80 | 0.9597 | 33.35 | 0.9158 | 32.04 | 0.8979 | 31.46 | 0.9221 | 38.25 | 0.9762 |
> > > | DDTB      | $\times$4               | 4/4       | 31.97   | 0.8920   | 28.46   | 0.7780   | 27.48   | 0.7330   | 25.77   | 0.7760   | --      | --       |
> > > | QuantSR-C | $\times$4               | 4/4       | 32.00 | 0.8924 | 28.50 | 0.7799 | 27.52 | 0.7342 | 25.88 | 0.7807 | 30.15 | 0.9040 |
> > >
> > > > **Q2b**: The performance of baselines for SRResNet provided in the paper and DDTB … is very inconsistent. Why is that?
> > >
> > > **A2b**: We clarify that we follow the results of full-precision 32-bit SRResNet reported in [3] because these recent results fully include the results on the Set5, Set14, B100, Urban100, and Manga109 datasets for fair comparison (while PAMS [4] and DDTB [2] missing Manga109). As we mentioned in Sec 4.1 of our paper, we follow the results and settings reported by existing work to the greatest extent, and the training settings for each quantized SR method are exactly the same to achieve a fair comparison.
> > >
> > > [3] Basic Binary Convolution Unit for Binarized Image Restoration Network. Xia, et al. ICLR,
> > >
> > > [4] PAMS: Quantized Super-Resolution via Parameterized Max Scale. Li, et al. ECCV, 2020.
> > >
> > > > **Q2c**: How did authors reproduce results of CADyQ with fixed bit width … ?
> > >
> > > **A2c**: As mentioned in our paper Sec 4.1, our implementation of CADyQ completely follows the official GitHub repository (Cheeun/CADyQ). To achieve the comparison on a fixed bit-width quantization, our key yet simple change is to make the three candidates in the search space of CADyQ the same (such as in the "train_edsrbaseline_cadyq.sh" file, under 8-bit settings --search_space is 8+8+8). Then we were able to achieve a fair comparison between different methods at the same bit width.

---

> > > > ### Comment · Reviewer_E4FQ · 2023-08-19
> > > >
> > > > Dear authors,
> > > >
> > > > Thanks for the clarifications and follow-up comments.
> > > > The concerns regarding efficiency and the performance of CADyQ are addressed.
> > > > One remaining and last concern is DDTB's reported results are obtained after training 60 epochs (according to the DDTB paper) while QuantSR is trained for 300 epochs. I don't think this is a fair comparison, as opposed to the authors' claim "the training settings for each quantized SR method are exactly the same to achieve a fair comparison".
> > > > Despite training for 5x more epochs, the PSNR difference is usually 0.02~0.03dB.
> > > > How is the performance of QuantSR when trained with the same setting (60 epochs and other details that may differ) as DDTB for fair comparison? (or DDTB's performance when trained with the same setting as QuantSR)
> > > >
> > > > Best,
> > > >
> > > > Reviewer E4FQ

---

> > > > > ### Author Response · Authors · 2023-08-19
> > > > > **We train our model with 26 epochs, which is much fewer than DDTB with 60 epochs**
> > > > >
> > > > > Dear Reviewer E4FQ,
> > > > >
> > > > > We thank the reviewer again for your time and feedback!
> > > > >
> > > > > We clarify that our QuantSR models are trained for **300K iterations** (as stated in L254 of our original manuscript), namely a total of **26 epochs** under the 32 batch size setting, instead of **300 epochs**. Therefore, although we directly compare the originally reported results of the DDTB paper using **60 epochs**, our QuantSR still achieves higher accuracy with **fewer training epochs**. And we thus believe that it is promising for our QuantSR to achieve better performance when increasing the training time (e.g., using the 60-epoch setting).
> > > > >
> > > > > We hope this clarification can solve your concerns. If you have any questions, please let us know. Thanks.
> > > > >
> > > > > Best,
> > > > >
> > > > > Authors

---

> > > > > > ### Comment · Reviewer_E4FQ · 2023-08-20
> > > > > >
> > > > > > Dear authors,
> > > > > >
> > > > > > Thanks for clarifying that the total number of training iterations for QuantSR is 300k iterations. I apologize for referring to wrong numbers. It is better to refer to the number of iterations (number of weight updates), in contrast to the number of epochs, which can differ depending on other training settings, such as batch size. Table 1 in [A] compares the number of training iterations for each seminal paper, where the total number of training iterations of DDTB is presented to be **3,000 iterations**, which is much fewer than that of QuantSR. Also, it's not just the number of training iterations; it seems other training details, such as batch size (32 vs 16), greatly differ.
> > > > > >
> > > > > > I understand the training setting in this paper is based on the setting from a recent work [B], and I'm not saying there is anything wrong with it. But, if the paper claims to outperform state-of-the-art algorithms, I believe the paper should compare against all state-of-the-art algorithms under the same setting, especially when settings, such as number of training iterations and batch size, have great impact on the final performance and when other works' results are reproduced under the same setting.
> > > > > >
> > > > > > I also understand that the authors may not be able to provide the reproduced results of DDTB as of now due to the limited amount of time. But, considering that DDTB paper has been published in 2022; the authors appear to be aware of DDTB before discussion phase; and the authors included the DDTB's reported results in the rebuttal, I believe the authors could have prepared reproduced results of DDTB in the original paper or been more careful with claim or presentation of results (such as writing footnote in the table saying that training settings differ) in the rebuttal and discussion phase.
> > > > > >
> > > > > > For the sake of completeness of the paper and preventing readers from being mislead in the future, I believe the authors could improve upon the presentations of results of previous works or tone down the claim if QuantSR does not outperform other state-of-the-art algorithms under the same setting.
> > > > > >
> > > > > > Please let me know if there is any misunderstanding.
> > > > > >
> > > > > > [A] Toward Accurate Post-Training Quantization for Image Super Resolution, CVPR2023
> > > > > >
> > > > > > [B] Basic Binary Convolution Unit for Binarized Image Restoration Network, ICLR 2023
> > > > > >
> > > > > > Best,
> > > > > >
> > > > > > Reviewer E4FQ

---

> > > > > > > ### Author Response · Authors · 2023-08-20
> > > > > > >
> > > > > > > Dear Reviewer E4FQ,
> > > > > > >
> > > > > > > We thank the reviewers for further feedback.
> > > > > > >
> > > > > > > We carefully checked the official DDTB repository (zysxmu/DDTB) and the original training logs provided (zysxmu/DDTB#trained-fp-models-and-quantized-models-here, downloaded "DDTB_bnsrresnetx2/4bit/"), here we compare the training of DDTB and QuantSR:
> > > > > > >
> > > > > > > **DDTB**: The training process contains two phases, the pre-training phase trains 500 epochs ("baseline.sh#L29") for SRResNet, and the quantization-aware training phase trains 60 epochs. Each epoch contains 1000 (1K) iterations (see "config.txt" and "log.txt"). The complete DDTB training thus costs (500+60)\*1K=**560K iterations**.
> > > > > > >
> > > > > > > **QuantSR**: There is no pre-training phase (training from scratch), and the complete quantization-aware training phase trains a total of **300K iterations**.
> > > > > > >
> > > > > > > The above comparison shows that our QuantSR uses fewer training iterations. Because DDTB requires statistics from the full-precision model (as the DDTB Initializer Section on page 9 of [1]), the complete training process must include a pre-training phase. While our QuantSR does not require additional pre-training.
> > > > > > >
> > > > > > > We are sorry again for missing DDTB results in the original manuscript, but we clarify that in both Table 2 of the attached PDF and Table A2a of the previous response, for the sake of rigor and the limited time, we directly followed and compared the results of DDTB [1] without any change. We will detail the settings for each quantization method as the reviewer suggested when adding these results to our next version.
> > > > > > >
> > > > > > > [1] Dynamic Dual Trainable Bounds for Ultra-low Precision Super-Resolution Networks. Zhong, et al. ECCV, 2022.
> > > > > > >
> > > > > > > Best,
> > > > > > >
> > > > > > > Authors

---

> > > > > > > > ### Comment · Reviewer_E4FQ · 2023-08-20
> > > > > > > >
> > > > > > > > Dear authors,
> > > > > > > >
> > > > > > > > I understand that DDTB has its special pre-training phase, but would it be really fair to count number of iterations during pre-training phase, when it's trained from scratch? Doesn't the number of iterations during quantization-aware training phase affect the final performance more? What about batch size? And what about other training details?
> > > > > > > >
> > > > > > > > The paper would be a great reference for this field and readers if the authors could include discussions on this and training details for previous works if they differ, similar to how it's done in the paper mentioned as [A] in my previous response.
> > > > > > > >
> > > > > > > > And, I'm ok with the acceptance of the paper even if the proposed method does not achieve state-of-the-art performance, since the performance is still great and the performance is not the only contribution.
> > > > > > > >
> > > > > > > > It's just that if I were authors, I would be more careful with presentation (visualizations regarding weight homogeneity, how previous works' results are presented, and exhaustive list of prior works), wordings of claims (SOTA performance), and sound evidence for claims (visualizations regarding weight homogeneity, latency, fair comparisons).
> > > > > > > >
> > > > > > > > Although I'm still a bit uncertain about comparisons against DDTB, the authors have addressed most of my concerns.
> > > > > > > > If the authors include all the discussions in the final version as promised, I'm ok with the acceptance.
> > > > > > > > Thus, I will raise my score.
> > > > > > > >
> > > > > > > > Best,
> > > > > > > >
> > > > > > > > Reviewer E4FQ

---

> > > > > > > > > ### Author Response · Authors · 2023-08-21
> > > > > > > > >
> > > > > > > > > Dear Reviewer E4FQ,
> > > > > > > > >
> > > > > > > > > We deeply appreciate your feedback and support! We will follow your suggestion to clarify the training and other details of quantization methods carefully and include all the discussions in the final version.
> > > > > > > > >
> > > > > > > > > Best,
> > > > > > > > >
> > > > > > > > > Authors

---

> > > ### Comment · Area_Chair_zHAx · 2023-08-19
> > >
> > > Dear Reviewer E4FQ,
> > >
> > > The authors have now submitted their further rebuttal to your second round of comments. Have they resolved your concerns?
> > >
> > > AC

---

### Official Review · Reviewer_G4U2 · 2023-07-01

**Soundness:** 3 good
**Presentation:** 3 good
**Contribution:** 3 good
**Rating:** 7
**Confidence:** 5

**Summary:**

This paper propose a new quantization method for single image super resolution, including Redistribution-driven Learnable Quantizer (RLQ) and Depth-dynamic Quantized Architecture (DQA). The previous one diversifies the representation and gradient information of quantized values by redistribution in quantizers, the last one improve the performance of quantized SR and achieves resource adaptation in inference. Extensive experiments demonstate the superiority of QuantSR.

**Strengths:**

The paper is well-written and easy to follow. The analysis of quantization for performance degradation is quite motivating and the proposed method is well-designed based on the analysis. The experiments show that QuantSR is better than previous methods.

**Weaknesses:**

1. The two main reasons for this performance degradation is much common, and not specific to super resolution task. Thus the method proposed in this paper is not SR-optimal.

2. Depth-dynamic Quantized Architecture seems not much related with quantization, which is unneccessary.

3. The designing of mapping method in Equation~5 is unclear, and is much similar with previous quantization method for image classifacation.

4. Learnable quantizer and soft gradient is not new for quantization.

5. Loss the latest baseline, DDTB[3].

[1] Zhou S, Wu Y, Ni Z, et al. Dorefa-net: Training low bitwidth convolutional neural networks with low bitwidth gradients[J]. arXiv preprint arXiv:1606.06160, 2016.

[2] Yamamoto K. Learnable companding quantization for accurate low-bit neural networks[C]//Proceedings of the IEEE/CVF conference on computer vision and pattern recognition. 2021: 5029-5038.

[3] Zhong Y, Lin M, Li X, et al. Dynamic dual trainable bounds for ultra-low precision super-resolution networks[C]//European Conference on Computer Vision. Cham: Springer Nature Switzerland, 2022: 1-18.

**Questions:**

1. what is the theory of mapping method in Equation~5, or which function that inspires you?

2. The true reason that cause the degradation is the high-dynamic activation [1][2], Can QuantSR solve this problem as well?

3. Depth-dynamic Quantized Architecture is similar with dynamic network, could show more comparison with previous methods.


[1] Tu Z, Hu J, Chen H, et al. Toward Accurate Post-Training Quantization for Image Super Resolution[C]//Proceedings of the IEEE/CVF Conference on Computer Vision and Pattern Recognition. 2023: 5856-5865.

[2] Zhong Y, Lin M, Li X, et al. Dynamic dual trainable bounds for ultra-low precision super-resolution networks[C]//European Conference on Computer Vision. Cham: Springer Nature Switzerland, 2022: 1-18.

**Limitations:**

Th authors addressed the limitations in this paper.

---

> ### Author Rebuttal · Authors · 2023-08-09
>
> > **Q1**: The two main reasons for this performance degradation is much common, ... Thus the method proposed in this paper is not SR-optimal.
>
> **A1**: We clarify that in this work, the techniques in QuantSR focus on the SR task since it possesses certain attributes not found in other high-level tasks, such as the retention of pixel-level representations, making the quantizing SR models challenging. However, despite our techniques being designed with SR in mind, they also have the potential to be applicable to other tasks. We intend to investigate this in future work.
>
> > **Q2**: DQA seems not much related with quantization, which is unneccessary.
>
> **A2**: We clarify that the DQA in QuantSR focuses on improving the accuracy of the quantized SR network by breaking the accuracy limit of their full-precision counterparts.
>
> As we discussed in Sec 3.1, in previous methods, the full-precision counterpart is commonly assumed to represent the performance upper limit of the quantized model. Our DQA breaks through this limitation through dynamic architecture and training strategies. By applying selective quantized blocks from the 2x-size architecture, the performance upper limit of QuantSR far surpasses that of directly training the same architecture. The weight sharing among variants also enables a versatile trade-off between accuracy and efficiency during inference.
>
> > **Q3**: The designing of mapping method in Equation~5 is unclear, and is much similar with previous quantization method for image classifacation.
>
> **A3**: As mentioned in Sec 3.2, the purpose of $\phi$ is to incorporate information reflecting the quantizer's actual behavior while ensuring optimization stability. In Eq. (5), our $\phi$ function is obtained by applying the $tanh(x)$ transformation and embedding it within each quantization interval. Specifically, in the interval $x \in [0, 1]$, the function is $\phi(x) = \frac{tanh(2x−1)}{tanh(1)}+0.5$, and the shape is repeated in other intervals.
>
> Compared to existing quantization methods that use soft functions, such as the tanh-based DSQ [1] and quadratic function [2], the $\phi$ function in RLQ has a stable impact in once backward while the former two are more intense. This property allows the quantized SR model to update stably without collapsing and ultimately enhances the accuracy by introducing forward vectorized information into the gradients. The results in *Table 2 in the attached PDF* demonstrate that QuantSR significantly outperforms other SR model quantized soft-function-based methods.
>
> [1] Gong R, et al. Differentiable soft quantization: Bridging full-precision and low-bit neural networks. ICCV, 2019.
>
> [2] Liu Z, et al. Bi-real net: Enhancing the performance of 1-bit cnns with improved representational capability and advanced training algorithm. ECCV, 2018.
>
> > **Q4**: Learnable quantizer and soft gradient is not new for quantization.
>
> **A4**: Please note that the techniques in QuantSR are specifically designed to address the performance degradation of the quantized SR model and are different from existing quantization methods, as described next.
>
> For the learnable quantizer: Since the SR task focuses on reconstructing images at the pixel level, making it crucial to preserve as much pixel-level information as possible during quantization. Previous quantized SR models using statistical quantizers lead to inaccurate information and significantly degraded precision. The forward mapping of our RLQ allows for flexible adjustments and diversification across the network, achieved by utilizing learnable intervals and zeros in quantizers.
>
> For soft gradient: As we respond in A3, the purpose of $\phi$ is to incorporate information reflecting the actual behavior of quantizers while ensuring optimization stability, and it makes QuantSR significantly outperform existing quantization methods using soft gradient approximation.
>
> > **Q5**: Loss the latest baseline, DDTB[3].
>
> **A5**: We will compare DDTB in the paper as the reviewer suggests. As we demonstrate in  *Table 2 of the attached PDF*, QuantSR consistently outperforms the DDTB method on various datasets.
>
> > **Q6**: what is the theory of mapping method in Equation~5, or which function that inspires you?
>
> **A6**: Regarding the mapping method of Eq. (5) in RLQ, it offers valuable gradient-guided information. Specifically, within each quantization interval, the gradient becomes smaller as the distance from the center of the region increases (L181 in our paper). This characteristic allows RLQ to incorporate information that accurately reflects the behavior of quantizers while maintaining optimization stability. Furthermore, in our response A3 above, we conduct a comparative analysis between RLQ and other existing soft quantization methods.
>
> > **Q7**: The true reason that cause the degradation is the high-dynamic activation [1][2], Can QuantSR solve this problem as well?
>
> **A7**: In our QuantSR, the learnable parameters in RLQ effectively tackle the quantization problem of high dynamic activations in SR networks. Relevant discussions on this approach will be included in the paper.
>
> Since the interval and zero-point of RLQ are learnable, it can gradually adapt to diverse and asymmetric input distributions. This implies that it can handle high dynamic activations better than statistics-based quantizers. As shown in Fig. 6(b), with increasing training, the offset of the activation quantizer in the network becomes more diverse, confirming the activation in the SR network is highly dynamic and demonstrating the ability of RLQ to adapt and cope well with such cases during training.
>
> > **Q8**: DQA is similar with dynamic network, could show more comparison with previous methods.
>
> **A8**: Our DQA is devoted to utilizing a dynamic architecture to break through the accuracy limitation of quantized models and also achieves dynamic trade-offs during inference. We will incorporate more related work about dynamic networks in the final version.

---

> > ### Comment · Reviewer_G4U2 · 2023-08-16
> > **Response to the rebuttal.**
> >
> > Thank you for the clear and convincing rebuttal. All my concerns are well addressed. This work is good enough to be published, I have raised my score.

---

### Official Review · Reviewer_VLLV · 2023-07-06

**Soundness:** 3 good
**Presentation:** 3 good
**Contribution:** 2 fair
**Rating:** 7
**Confidence:** 4

**Summary:**

This paper propose an low-bit quantization method for efficient image super-resolution by introduce a Redistribution-driven Learnable Quantizer (RLQ). Besides, the proposed Depth-dynamic Quantized Architecture (DQA) allows for the trade-off between efficiency and accuracy during inference through weight sharing. Experiments show that the proposed method outperforms existing state-of-the-art quantized SR networks in terms of accuracy while also providing more competitive computational efficiency.

**Strengths:**

- **Originality:** This paper propose a novel accurate quantization scheme for efficient image SR by introducing Redistribution-driven Learnable Quantizer (RLQ) and Depth-dynamic Quantized Architecture (DQA). The proposed scheme can be applied into both CNN- and Transformer- based SR networks.

- **Quality:** This paper is well organized and easy to follow.

**Weaknesses:**

- Motivation is unclear: Although the proposed method outperforms existing state-of-the-art quantized SR networks in terms of trade-off between accuracy and complexity, the contribution is still limited. If we only look at the method section, it seems that the proposed module can be applied to Low-bit Quantization of model compression for various tasks. Are there any special designs for SR tasks? If so, it is suggested to provide more discussion to highlight the contributions.

- As claimed in Introduction section, "existing SR models rely on expensive computational resources, which significantly limits the real-world SR applications on resource-constrained edge devices. Therefore, there is an urgent requirement to develop model compression techniques for SR models to reduce the computational overhead". As far as I know, the low complexity SR model is indeed an important research task, but the model compression is only one of them. In fact, many lightweight SR algorithms have been proposed recently, such as look-up table based method, *e.g.,* SR-LUT (CVPR2021), MuLUT(ECCV2022). It is suggested to discuss the superior of the proposed method.

- The proposed method is evaluated only on two classical SR networks, SRResNet (CVPR'17) for CNN-based and SwinIR_S (ICCVW'21) for Transform-based backbones, respectively. Can the proposed method be extended to the recently published SR models?

**Questions:**

Please refer to comments in the **weakness** part.

---

> ### Author Rebuttal · Authors · 2023-08-09
>
> > **Q1**: … Are there any special designs for SR tasks? If so, it is suggested to provide more discussion to highlight the contributions.
>
> **A1**: We clarify that in this work, the techniques in QuantSR focus on the SR task since it possesses certain attributes not found in other high-level tasks, such as the retention of pixel-level representations, making the quantizing SR models challenging. However, despite our techniques being designed with SR in mind, they also have the potential to be applicable to other tasks. We intend to investigate this in future work.
>
> Specifically, the Redistribution-driven Learnable Quantizer (RLQ) targets the preservation of fine-grained pixel information essential for SR tasks. The information recovery of RLQ for quantized SR networks is accomplished through forward and backward propagation by utilizing learnable quantizer parameters and the embedded soft function, respectively. And the Depth-dynamic Quantized Architecture (DQA) leverages the blockwise characteristics of the quantized SR model, effectively pushing the boundaries of quantization performance. Benefiting from screening from a stronger initial SR model with more stacked quantized blocks, the upper limit of the model accuracy is shattered.
>
> Both RLQ and DQA operate independently but synergistically, collectively enhancing the performance of quantized SR models. In addition, we keep the general nature of the proposed techniques, which are applicable to a variety of SR architectures, including CNNs and transformers.
>
> > **Q2**: … many lightweight SR algorithms have been proposed recently, such as look-up table based method, e.g., SR-LUT (CVPR2021), MuLUT(ECCV2022). It is suggested to discuss the superior of the proposed method.
>
> **A2**: We will follow the reviewer's suggestions to discuss the mentioned works. Currently, numerous researchers are devoted to achieving lightweight quantization algorithms, such as SR-LUT and MuLUT, which innovatively utilize lookup table-based methods to achieve fast inference and good reasoning speed. However, they still exhibit a significant gap in accuracy compared to advanced SR models based on deep networks. For instance,the PSNR indicator for the 4x Set5 dataset: MuLUT-SDY-X2 30.60 vs. 4-bit QuantSR-T 32.18. Our proposed QuantSR approach aims to bridge the gap between quantized SR models and their full-precision versions, bringing its performance close to that of their full-precision counterparts and surpassing existing methods, including both quantized and lookup-table-based SR models. Additionally, since the quantizer in QuantSR works at a general operator level, it has the potential to further accelerate convolutional units in lookup-table-based SR models, maintaining performance while pushing their efficiency limits further.
>
> > **Q3**: … Can the proposed method be extended to the recently published SR models?
>
> **A3**: Yes, thanks to the general nature of our QuantSR, it can be flexibly applied to various types of SR models. In this study, we utilize our QuantSR to quantize the CAT model under 4-bit and present the relevant results in Table A3 (part of *Table 1 in the attached PDF*). The findings demonstrate that QuantSR significantly enhances the model’s performance compared to PAMS, the currently existing state-of-the-art quantized SR model. This outcome illustrates the robust improvements our proposed method brings to the quantization of various SR architectures.
>
> Table A3: Quantitative results. CAT-R [1] is used as full-precision backbones.
> |                             |                         | \#Bit   | Set5 |  | Set14 | | B100 | | Urban100 | | Manga109 | |
> |-----------------------------|-------------------------|-----------|-------|--------|-------|--------|-------|--------|-------|--------|-------|-------------|
> | Method    | Scale | ($w$/$a$) | PSNR  | SSIM   | PSNR  | SSIM   | PSNR  | SSIM   | PSNR  | SSIM   | PSNR  | SSIM |
> | Bicubic                     | $	\times$4                | -/-       | 28.42 | 0.8104 | 26.00 | 0.7027 | 25.96 | 0.6675 | 23.14 | 0.6577 | 24.89 | 0.7866      |
> | CAT\_R | $\times$4               | 32/32     | 32.89 | 0.9044 | 29.13 | 0.7955 | 27.95 | 0.7500 | 27.62 | 0.8292 | 32.16 | 0.9269      |
> | PAMS                        | $\times$4               | 4/4       | 31.77 | 0.8885 | 28.33 | 0.7752 | 27.40 | 0.7295 | 25.55 | 0.7672 | 29.62 | 0.8966      |
> | QuantSR-T                   | $\times$4               | 4/4       |  32.31 | 0.8955 | 28.74 | 0.7836 | 27.61 | 0.7389 | 26.31 | 0.7896 | 30.69 | 0.9118     |
>
> [1] Chen et al. Cross Aggregation Transformer for Image Restoration. NeurIPS, 2022.

---

> > ### Comment · Reviewer_VLLV · 2023-08-19
> >
> > Thank you for the clear and convincing rebuttal. All my concerns are well addressed. I will raised my score.

---

### Author Rebuttal · Authors · 2023-08-09

We appreciate all reviewers for the constructive reviews and positive feedback to our QuantSR. Your expertise and insightful comments help us to further improve our paper.

The attached PDF includes:

* Figure 1: Visual comparison of weights in QuantSR with and without learnable quantizer parameters.

* Table 1: Quantitative results on transformer-based SwinIR-S and CAT-R architectures (4-bit, 4$\times$ scale)

* Table 2: Quantitative results on CNN-based SRResNet architectures (4-bit, 4$\times$ scale)

For detailed instructions, please see our responses to each reviewer.

---

### Decision · Program_Chairs · 2023-09-21

**Decision:**

Accept (spotlight)

**Comment:**

Before the discussion and rebuttal, reviewers highlighted the following strengths and weaknesses of Submission 422:

*Strengths:*
- The paper introduces a novel quantization scheme for efficient image super-resolution (SR) using Redistribution-driven Learnable Quantizer (RLQ) and Depth-dynamic Quantized Architecture (DQA). This scheme can be applied to both CNN and Transformer-based SR networks.
- The paper is well-organized, easy to follow, and well-written.
- The proposed RLQ addresses weight homogeneity issues without additional inference overhead, and DQA effectively manages the performance-efficiency trade-off.
- Experiments demonstrate that the proposed method, QuantSR, outperforms existing quantized SR networks in terms of accuracy, and the results show significant performance improvement across various bit widths.
- The paper offers a combination of learnable structured pruning with learnable low-bit quantization, which is an innovative attempt.

*Weaknesses:*
- The motivation for the proposed method is unclear, and the unique designs tailored for SR tasks are not highlighted sufficiently.
- The paper does not sufficiently discuss or compare its contributions against existing lightweight SR algorithms or other model compression techniques.
- There is limited evaluation on recently published SR models, raising concerns about the method's generalizability.
- Some reviewers found aspects of the proposed RLQ and DQA methods, such as the function \phi and its role in forward propagation, to be ambiguous or unclear.
- The paper lacks a demonstration of the extent of weight homogeneity before and after the proposed quantization scheme.

Post-rebuttal, reviewers appreciated the clear and convincing responses provided by the authors. They acknowledged that most concerns were well addressed, leading to raised scores. While one reviewer still had reservations about certain comparisons, they recognized the paper's value and agreed with its acceptance. The decision is to accept the paper.